# TIFO: TIME-INVARIANT FREQUENCY OPERATOR FOR STATIONARITY-AWARE REPRESENTATION LEARNING IN TIME SERIES

## ABSTRACT

Nonstationary time series forecasting suffers from the distribution shift issue due to the different distributions that produce the training and test data. Existing methods attempt to alleviate the dependence by, e.g., removing low-order moments from each individual sample. These solutions fail to capture the underlying time-evolving structure across samples and do not model the complex time structure. In this paper, we aim to address the distribution shift in the frequency space by considering all possible time structures. To this end, we propose a Time-Invariant Frequency Operator (TIFO), which learns stationarity-aware weights over the frequency spectrum across the entire dataset. The weight representation highlights stationary frequency components while suppressing non-stationary ones, thereby mitigating the distribution shift issue in time series. To justify our method, we show that the Fourier transform of time series data implicitly induces eigen-decomposition in the frequency space. TIFO is a plug-and-play approach that can be seamlessly integrated into various forecasting models. Experiments demonstrate our method achieves 18 top-1 and 6 top-2 results out of 28 forecasting settings. Notably, it yields 33.3% and 55.3% improvements in average MSE on the ETTm2 dataset. In addition, TIFO reduces computational costs by 60% -70% compared to baseline methods, demonstrating strong scalability across diverse forecasting models. Our code can be found at this anonymous GitHub repository[1].

## 1 INTRODUCTION

Time series forecasting is vital to decision-making in real-world applications like industrial system control and stock market tracking. However, a crucial challenge is the non-stationary nature of real-world time series that often leads to poor generalization to unseen data beyond the training set, known as *distribution shift issue in time series*. Consequently, the non-stationary characteristics of time series data necessitate the development of forecasting models that are robust to such temporal shifts in data distribution, while failing to address this challenge often leads to representation degradation and compromised model generalization In this paper, we analyze this issue and existing normalization-based solutions (Kim et al., 2021; Fan et al., 2023; Liu et al., 2023; Han et al., 2024) from a data generation perspective. We thus introduce a principled new solution derived from this analysis.

From a distributional perspective, a time series is sampled from a distribution $x \sim p(x|t)$, where $t$ denotes a temporal condition (e.g., the $t$-th sliding window) drawn from a time-evolving distribution $p(t)$. Consider the normal distribution $\mathcal{N}(\mu_t, \sigma_t^2)$ for example, it suggests that the mean and variance are conditional on $t$. Therefore, a forecasting model trained on training data $x_{\text{train}} \sim p(x|t_{\text{train}})$ may not perform well on test data $x_{\text{test}} \sim p(x|t_{\text{test}})$ since $t_{\text{train}}$, $t_{\text{test}}$ can be vastly different, referred to as the distributional shift issue.

Existing methods tackle this issue by weakening the dependency of $x$ on $t$ through normalizing the data distribution (Kim et al., 2021; Liu et al., 2023; Fan et al., 2023; Han et al., 2024), so that the time-dependent low-order moments (mean and variance) are removed from both training and test sets to obtain a *standard distribution*, in the normal case $\mathcal{N}(0, 1)$. While this kind of method has

---

[1] https://anonymous.4open.science/r/TIFO-6BE1

shown some promise, it implicitly assumes that (1) this standard reference distribution represents the underlying distribution of the entire dataset, and (2) low-order statistics are sufficient to describe the complex data distribution and avoids modeling the time parameter distribution $p(t)$. It may cause poor performance when the assumption does not hold, e.g., the distribution has more complex dependency over time, such as modality, high-order moments, or its functional form. To address this challenge, this paper proposes a novel frequency-based method and provides its theoretical foundation.

From a signal processing perspective, non-stationarity in real-world time series often manifests as changes in frequency characteristics (Proakis & Manolakis, 1996), such as shifts in dominant spectral modes or time-dependent amplitudes. Mean and variance characterize the overall amplitude and spread of a time series, but they fail to capture how energy is distributed across different frequency components (Piao et al., 2024). In the existing works, normalizing low-order statistics may help align total energy but not its spectral structure, such as the location of that energy in the frequency space. As a result, frequency shifts (i.e., changes in the dominant frequencies over time) may persist, especially when spectral characteristics differ significantly across training and testing datasets.

In this work, we propose to address distributional shift by working in the frequency space and by considering all possible time conditions via $p(x) = \int p(t)p(x|t)\mathrm{d}t$. Specifically, frequency-domain analysis provides a disentangled view of underlying temporal features, enabling the model to capture fine-grained stationarity. Crucially, such analysis is conducted across samples at the dataset level: the observed distribution $p(x)$ is formulated as a weighted average of the conditional distributions $p(x|t)$ over all possible time conditions, where the weights are given by $p(t)$. Thus, we can achieve the same goal (weakening the dependency of $x$ on $t$) but account for the full temporal variability. To this end, we propose Time-Invariant Frequency Operator (TIFO) for stationarity-aware representation learning, which consistes two stages. **Stage-I**: We apply the Discrete Fourier Transform (DFT) to all samples in a given time series dataset to obtain their frequency components. For each frequency, we then conduct cross-sample statistical analysis and use a lightweight neural network layer to learn a weight that quantifies its time-invariant relevance for mitigating distributional shift. Through this data-driven weighting, our method emphasizes relatively stationary components (via higher weights) while suppressing non-stationary ones (via lower weights), effectively learning a weighted average over all time conditions embedded in the dataset. **Stage-II**: After weighting, we perform an inverse DFT (IDFT) to project the adjusted frequencies back into the time domain. These transformed time series are then fed into forecasting models. The weighted composition of fine-grained frequency components enables the model to approximate more complex, temporally-evolving distributions. Moreover, this paper takes a first step toward providing a theoretical foundation to justify our method. We adopt a non-stationary stochastic process perspective and characterize time series through their frequency characteristics. We show that by classical harmonic analysis results, the Fourier transform on time series data implicitly induces a kernel in the frequency space, which in turn permits a set of orthonormal basis functions formed by spectral eigen-decomposition (Berg et al., 1984). By learning data-specific eigenvalues, the frequency components that are responsible for distributional discrepancies can be captured as a weighted sum of eigenfunctions. Our contributions are as follows:

> **Summary of Key Contribution**
>
> **(Perspective)** We provide a data generation-based formulation of non-stationary time series and distributional shift, offering a unified theoretical framework that both explains and generalizes existing normalization methods.
>
> **(Method)** We propose to learn stationarity and non-stationarity across samples in the frequency domain. Our method enables fine-grained feature extraction to handle complex temporal dynamics, and can be seamlessly integrated into various forecasting models. We also provide a theoretical analysis to justify the soundness of our method.
>
> **(Experiments)** We apply TIFO to popular forecasting models, including DLinear (Zeng et al., 2023), PatchTST (Nie et al., 2023), and iTransformer (Liu et al., 2024b) to validate its effectiveness across seven datasets. In non-stationary datasets such as ETTm2, we improve PatchTST and iTransformer by 33.3% and 55.3%, respectively. Compared to existing normalization methods, TIFO achieves 18 top-1 and 6 top-2 results out of 28 settings. Analysis on data distribution shows that TIFO reduces the difference between training and testing datasets by up to 88%, improving robust forecasting for non-stationary data. Computational efficiency analysis shows that TIFO achieves improvements of 60% to 70% in 16 out of 28 settings.

## 2 BACKGROUND AND PRELIMINARY ANALYSIS

We consider multivariate time series forecasting, where we are given a set of input $\mathcal{X} = \{\mathbf{X}^{(i)}\}_{i=1}^{N}$ and the corresponding target $\mathcal{Y} = \{\mathbf{Y}^{(i)}\}_{i=1}^{N}$ in discrete time, where $N$ denotes the number of sequences. Let $C, L_x, L_y$ respectively denote the number of variables, the input-sequence length, and the model prediction length, then the goal can be formulated as that given an input sequence $\mathbf{X}^{(i)} \in \mathbb{R}^{L_x \times C}$, predict the target values $\mathbf{Y}^{(i)} \in \mathbb{R}^{L_y \times C}$.

### 2.1 ANALYZE DISTRIBUTIONAL SHIFT FROM A DATA GENERATION PERSPECTIVE

In this paper, we tackle the distributional shift issue in time series forecasting by analyzing the data generation process. Time $t$ can be viewed as the index of a structured temporal sequence (e.g., the $t$-th sampling window) drawn from a distribution $p(t)$. Once $t$ is sampled, a corresponding time series segment $\mathbf{X}$ is then drawn from the conditional distribution $p(x|t)$ (Adak, 1998), reflecting various events (e.g., industrial sensors) occurring within that segment. This formulation highlights that time series datasets $\mathcal{X} = \{\mathbf{X}^{(i)}\}_{i=1}^{N}$ can be viewed as being generated from different realizations of temporal indices $\{t^{(i)}\}_{i=1}^{N}$, where each $t$ induces its own conditional distribution $p(x|t)$. As time evolves, these context-dependent distributions naturally shift, reflecting the non-stationary characteristics of time series. As such, the distributional difference of $p(x|t_{\text{train}}) \neq p(x|t_{\text{test}})$ thus arises from the underlying variation in temporal contexts. In practice, this shift is often quite large, since testing or future time series naturally change with time and differ from the training contexts.

### 2.2 NORMALIZATION METHODS WEAKEN $x$ DEPENDENCY ON TIME CONDITION $t$

Methods such as RevIN (Kim et al., 2021) and SAN (Liu et al., 2023) are based on a key concept: they aim to estimate time-dependent statistics (e.g., mean and variance) and remove them from the input time series. This process reduces the conditional dependence on $t$, transforming the time-varying distribution $p(x|t)$ closer to a stationary form $p(x)$. We provide more discussion in Appendix B.1.

Formally, take a Gaussian distribution $\mathcal{N}(\mu_t, \sigma_t^2)$ for example, its mean and standard deviation are subject to time changes. They estimate $(\mu_t, \sigma_t)$ for each time segment $x_t$ via a function $f_\theta(x_t)$, where $f_\theta(\cdot)$ can be either a numerical computation (Kim et al., 2021) or a neural network (Fan et al., 2023; Liu et al., 2023; Han et al., 2024). By removing these statistics from the data via $\hat{x}_t = \frac{x_t - \mu_t}{\sigma_t}$, the time-dependent $\mathcal{N}(\mu_t, \sigma_t^2)$ is transformed into a standard Gaussian $\mathcal{N}(0, 1)$. Consequently, the distributional shift between training and test datasets is mitigated, since $\hat{x}_{\text{train}}, \hat{x}_{\text{test}} \sim \mathcal{N}(0, 1)$.

Nonetheless, some limitations remain in existing normalization methods:

1. **Inadequate Data Distribution Modeling.** This approach handles each sample individually that implicitly assumes the reference $\mathcal{N}(0, 1)$ as the dataset-level ground truth distribution, i.e., $\forall t \in \{\text{train}, \text{test}\}, \ p(\hat{x}_t) = \mathcal{N}(0, 1)$. This suppresses meaningful cross-sample stationaries and prevents the model from capturing how data evolves globally across training and test domains.

2. **Simplistic Distribution Characterization.** Existing methods primarily rely on low-order statistics $(\mu, \sigma)$ to standardize data, which assumes that the data distribution can be described by Gaussian-like behavior. However, real-world time series often exhibit far more complex characteristics (e.g., non-stationary frequency dynamics) and distributions (consider the Student's t distribution, for instance). Even after normalization, residual distributional shifts may persist.

### 2.3 PROBLEM FORMULATION

The above discussion suggests that distributional shift in time series should be understood in terms of how data evolves globally across multiple samples, and in identifying which characteristics can capture such evolution. The key question in this paper becomes:

**Problem 1.** *which aspects of the data distribution are consistent across multiple samples, and which are unstable and lead to shifts between training and testing domains?*

Let each sample be generated as $x \sim p(x|t)$ with $t \sim p(t)$ denoting a latent temporal condition. Similar to normalization methods, given a training dataset $\mathcal{X} = \{\mathbf{X}^{(i)}\}_{i=1}^{N}$, to mitigate the distribution

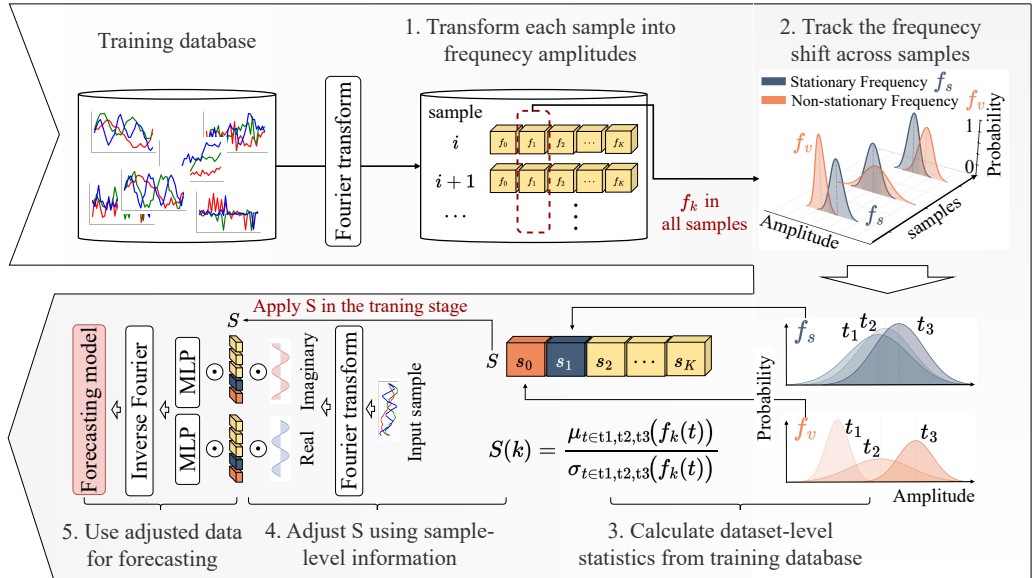

Figure 1: Overview of TIFO. Before training, we first transfer all samples into the frequency domain and measure their cross-sample stationarity at the dataset level (steps 1 & 2). These features are then used to learn frequency weights that measure frequency stationarity (step 3). During training, each input sample is transformed into the frequency domain and then weighted by the learned stationarity weights. Finally, they are transformed to the time domain to serve as input to the forecasting models (steps 4 & 5). TIFO is optimized using the forecasting loss along with the backbone model.

shift issue, the objective is to learn a transformation $f(\cdot)$ such that the resulting representations that learn more stationary features from and suppress the non-stationary ones. However, we argue that this should be satisfied with two requirements: emphasizes stationary components across all samples from $f(\mathcal{X})$ while preserving important characteristics unique to each sample $f(\mathbf{X}^{(i)})$, since time series data often contain stochastic variations and local structures (Piao et al., 2024). To this end, we develop a novel two-stage framework that mitigates distributional shift and is plug-and-play for diverse forecasting models, similar to existing methods (Kim et al., 2021; Ye et al., 2024; Han et al., 2024).

## 3 PROPOSED METHOD

In this section, we introduce our novel framework TIFO. We begin by introducing our design of TIFO in Section 3.1. This design is supported by a detailed theoretical analysis in Section 3.2.

### 3.1 TIFO: SYSTEM OVERVIEW AND FEEDFORWARD PIPELINE

Figure 1 shows the processing pipeline of TIFO. It includes a two-stage modeling: (i) pre-training: measuring frequency stationarity in the entire training set; (ii) in-training: adaptively re-weighting frequency coefficients of each input sequence.

**Stage-I: Dataset-level Stationarity Learning**

- *Step 1.* For each sample $\mathbf{X}^{(i)} \in \mathbb{R}^{L \times C}$ from a multi-channel time series training dataset $\mathcal{X} = \{\mathbf{X}^{(i)}\}_{i=1}^{N}$, we take the Discrete Fourier Transform (DFT) and obtain the amplitudes $\mathbf{A}^{(i)}(k,c)$ for frequency $k$ and channel $c$. Where $L$ represents the length of each sample, C is the number of channels, and $K$ is the number of frequency components.

- *Step 2.* We aggregate across the training set to measure the stationarity of frequencies using

$$\mathbf{S}(k,c) = \frac{\mu_{i \in \text{train}}(\mathbf{A}^{(i)}(k,c))}{\sigma_{i \in \text{train}}(\mathbf{A}^{(i)}(k,c))}$$

, where $\mu_{i\in\text{train}}(\cdot)$ and $\sigma_{i\in\text{train}}(\cdot)$ are the mean and standard deviation of the amplitude $\mathbf{A}^{(i)}(k, c)$ in all samples. A larger $\mu$ means a higher energy proportion, while a greater $\sigma$ denotes higher sample dispersion. Thus, a higher $\mathbf{S}(k, c)$ reflects more stationary frequency behavior.

- *Step 3.* In this step, we save the calculated $\mathbf{S}$ from Step 2 and pass it to the forecasting model. Through $\mathbf{S}$, the model can access cross-sample variation information, while the MLPs in Stage II can further optimize the weights through training. This aims to allow the forecasting model to learn the overall stationarity of each component across the dataset, and use this information to learn a stationary representation of data during the training.

**Stage-II: Sample-specific Learning & Forecasting**

- *Step 4.* In the training stage, given an input sample $\mathbf{X} \in \mathbb{R}^{L\times C}$, we compute its DFT and obtain real and imaginary coefficients $\mathbf{R}, \mathbf{I} \in \mathbb{R}^{K\times C}$. Based on the pre-computed stationarity $\mathbf{S} \in \mathbb{R}^{K\times C}$, we use two independent MLPs to generate frequency weights:

$$\lambda_r = \text{MLP}_r(\mathbf{S}), \quad \lambda_i = \text{MLP}_i(\mathbf{S}). \tag{1}$$

The weighted coefficients are then obtained by element-wise multiplication: $\mathbf{R}_w = \mathbf{R} \odot \lambda_r$, $\mathbf{I}_w = \mathbf{I} \odot \lambda_i$. We model the real and imaginary parts separately to ensure that when the weighted coefficients are mapped to the time domain through the inverse DFT, the resulting amplitudes remain non-negative, allowing the inverse DFT to output real-valued sequences. In this stage, TIFO serves as a lightweight frequency stationarity filter, enhancing stationary components (those with a high stationarity score) while suppressing non-stationary ones, thereby addressing the distribution shift problem defined in Section 2.

- *Step 5.* Finally, the weighted coefficients $(\mathbf{R}_w, \mathbf{I}_w)$ are transformed to the time domain via inverse DFT $\widetilde{\mathbf{X}} = \text{iDFT}(\mathbf{R}_w + i\,\mathbf{I}_w)$, and fed into the backbone forecasting models. $\widetilde{\mathbf{X}} \in \mathbb{R}^{H\times C}$ is the final output of TIFO and the input of the backbone model, where $H$ denotes the forecasting horizon. The whole framework is optimized end-to-end using the forecasting MSE loss.

## 3.2 THEORETICAL ANALYSIS

Our frequency weighting in the previous section is the result of the following theoretical analysis. This section connecting nonstationarity to spectrum analysis is novel to the best of our knowledge.

**Existence of Time-Averaged Representation.** We connect the learning of frequency weights Eq.(1) to spectrum analysis by noticing that these weights correspond eigenvalues that characterize frequency space representations that discern frequency components responsible for distributional shifts. We begin our analysis with the assumption:

**Assumption 1.** *The time series dataset $\mathcal{X} = \{\mathbf{X}^{(i)}\}_{i=1}^{N}$ is composed of multiple samples from $t_i \sim p(t)$ and $x^{(i)} \sim p(x|t_i)$ so it can sufficiently representation the distributions.*

We assume that the time series dataset can sufficiently represent the time variations. This assumption is realistic in many real-world datasets such as electricity or stock markets that collect data on many-year-basis. These datasets compose a challenge to normalization methods, since removing empirical estimates $\bar{\mu}, \bar{\sigma}$ from a batch does not equal removing $\mu_{t_i}, \sigma_{t_i}$ that is governed by a specific $t_i$. It is intractable to identify which data batch is governed by a unique time structure. Our method instead turns to *a time-averaged representation*. We integrate over time by applying Fourier transform on data. The next theorem formalizes this idea.

**Theorem 1** (Bochner's Theorem (Scholkopf & Smola, 2001))**.** *A kernel function $k(x, y) \geq 0$ is a distance measure of input $x, y$. It is valid if and only if there exists a probability density that is the Fourier transform of the kernel.*

The fact that we assume data is generated by $t_i \sim p(t)$, $x^{(i)} \sim p(x|t_i)$ plus we apply Fourier transform to $\mathcal{X}$ imply that a kernel function exists on the frequency domain:

$$k(\omega_1, \omega_2)|_x = \int_{\mathbb{R}} e^{it(\omega_1-\omega_2)} \mathcal{X} \mathrm{d}t = \mathbb{E}_t \left[ e^{it(\omega_1-\omega_2)} \right] \Big|_x. \tag{2}$$

where we use $|_x$ to denote the dependency on $x$. While in practice the dataset $\mathcal{X}$ needs to be infinitely large to sufficiently represent the distributions, we can expect that with a reasonably sized dataset that

comprises multiple samples of $t$, the existence of a kernel is guaranteed. As the result of integration over time, it is also reasonable to expect that the kernel as a time-averaged representation should perform better than the normalization methods.

**Adapting Time-Averaged Representation to Data.** The kernel function in Eq. (2) is implicit since we know it exists but have no access to it. To exert the kernel as a distance measure, we can adapt it to input data so the distances between important and unimportant frequency components are emphasized most. To this end, we explicitly learn the kernel in a data-driven way based on the Mercer's theorem.

**Theorem 2** (Mercer's Theorem (Mercer, 1909)). *A valid, positive definite kernel function can be represented by a set of eigenfunctions that form an orthonormal basis $\{\zeta_i\}_{i \in \mathbb{N}}$ with associated eigenvalues $\lambda_1 \geq \lambda_2 \geq \cdots > 0$ such that:*

$$k(\omega_1, \omega_2) = \sum_{i=1}^{\infty} \lambda_i \zeta_i(\omega_1) \zeta_i(\omega_2),$$

(3)

*where the convergence of the infinite series holds absolutely and uniformly.*

Because the kernel must exist in the frequency space, by Mercer's theorem it must permit the eigen-decomposition that forms a set of orthonormal basis in the space. Moreover, if we impose a structure on the eigenfunctions $\zeta$, then the kernel varies with the eigenvalues $\lambda$. Therefore, learning the eigenvalues given data is equivalent to learning the kernel itself (Wilson et al., 2016).

We follow (Xu et al., 2019) to employ the assumption that the kernel is periodic, which is natural for frequencies. Therefore, the kernel has the Fourier basis as its eigenfunctions: $\zeta_1(\omega) = 1$, $\zeta_{2j}(\omega) = \cos\left(\frac{2\pi j \omega}{t}\right), \zeta_{2j+1}(\omega) = \sin\left(\frac{2\pi j \omega}{t}\right)$ for $j = 1, 2, \ldots$. Now $\lambda_i, i = 1, 2, \ldots$ become the corresponding Fourier coefficients to weight the contribution of each $\zeta$. Therefore, we have concluded the theoretical analysis on the role played by $\lambda$ introduced in Eq.(1).

## 4 EXPERIMENTS

In this section, we conduct experiments to answer the following research questions: *RQ1. Forecasting Accuracy.* Does TIFO improve forecasting performance on non-stationary datasets? *RQ2. Addressing Distribution Shift.* Does learning $\lambda$ mitigate the distribution shift? *RQ3. Frequency Feature Learning.* How do $\lambda$ affect the backbone models to capture informative frequency characteristics? We first introduce the experimental datasets and settings, followed by detailed results and analysis to answer each of the above questions. We also conduct efficiency analysis and ablation studies.

### 4.1 EXPERIMENT SETTINGS

**Datasets.** We benchmark our models on seven widely used multivariate time-series datasets: Electricity Transformer Temperature (ETT) with four subsets at hourly (ETTh1, ETTh2) and 15-minute (ETTm1, ETTm2) resolutions; Electricity consumption of 321 clients; Traffic volumes from 862 San Francisco sensors; and Weather recordings of 21 meteorological variables. We follow the Time–Series-Library split (7:2:1) with a fixed window length $L = 96$ and apply per-channel $z$-score normalisation; this rescales variables but leaves cross-instance non-stationarity intact. Models are trained with the Mean-Squared-Error loss and evaluated in the time domain by MSE and MAE.

**Baselines.** We selected RevIN (Kim et al., 2021), SAN (Liu et al., 2023), and FAN (Ye et al., 2024) as our baselines. RevIN is widely used as a fundamental module in various forecasting models, including PatchTST (Nie et al., 2023), iTransformer (Liu et al., 2024b), among others (Wang et al., 2024). SAN is a normalization-based method that outperforms several non-stationary forecasting modules (Kim et al., 2021; Fan et al., 2023). We also selected FAN, it introduces a frequency-domain modeling normalization-based method to address the distributional shift issue.

**Backbones and Setup.** For fair comparisons, we selected three forecasting models, including DLinear (Zeng et al., 2023), PatchTST (Nie et al., 2023), and iTransformer (Liu et al., 2024b), as the backbones, and deployed all non-stationary modules (TIFO, RevIN, SAN, and FAN) for evaluation. DLinear is a simple yet efficient forecasting model with an architecture solely involving MLPs. PatchTST and iTransformer are two well-known Transformer methods that frequently serve

as baselines in various forecasting research (Liu et al., 2024b;a; Piao et al., 2024; Zhang et al., 2024b). We followed the implementation and setup provided in (Liu et al., 2023) and (Liu et al., 2024b).

**Experiments Details.** All experiments were implemented on a single NVIDIA RTX A6000 48GB GPU. More details of the datasets are in Appendix B.4, the preprocessing are in B.5, the baselines are in B.6, the backbones and setup are in B.7, and other details of the experiments are in B.8.

## 4.2 EXPERIMENT RESULTS

Table 1: Multivariate forecasting results (average) with forecasting lengths $H \in \{96, 192, 336, 720\}$ for all datasets and fixed input sequence length $L = 96$.

| Models | PatchTST (Nie et al., 2023) | | | | iTransformer (Liu et al., 2024b) | | | |
| Methods | + TIFO | | Ori | | + TIFO | | Ori | |
| Metric | MSE | MAE | MSE | MAE | MSE | MAE | MSE | MAE |
|---|---|---|---|---|---|---|---|---|
| ETTh1 | **0.438** ± 0.024 | **0.437** ± 0.035 | 0.480 ± 0.037 | 0.481 ± 0.031 | **0.445** ± 0.017 | **0.443** ± 0.026 | 0.511 ± 0.033 | 0.496 ± 0.036 |
| ETTh2 | **0.379** ± 0.032 | **0.380** ± 0.038 | 0.604 ± 0.130 | 0.524 ± 0.027 | **0.376** ± 0.041 | **0.400** ± 0.057 | 0.813 ± 0.134 | 0.666 ± 0.072 |
| ETTm1 | **0.390** ± 0.027 | **0.398** ± 0.025 | 0.419 ± 0.055 | 0.432 ± 0.047 | **0.396** ± 0.026 | **0.406** ± 0.056 | 0.447 ± 0.026 | 0.457 ± 0.061 |
| ETTm2 | **0.280** ± 0.032 | **0.325** ± 0.031 | 0.420 ± 0.035 | 0.424 ± 0.044 | **0.283** ± 0.020 | **0.327** ± 0.026 | 0.633 ± 0.055 | 0.489 ± 0.041 |
| Electricity | **0.197** ± 0.027 | **0.296** ± 0.033 | 0.218 ± 0.31 | 0.307 ± 0.032 | **0.169** ± 0.035 | **0.262** ± 0.041 | 0.179 ± 0.028 | 0.279 ± 0.046 |
| Traffic | **0.427** ± 0.029 | **0.285** ± 0.025 | 0.619 ± 0.077 | 0.365 ± 0.029 | **0.424** ± 0.031 | **0.282** ± 0.027 | 0.576 ± 0.069 | 0.372 ± 0.035 |
| Weather | **0.251** ± 0.019 | **0.276** ± 0.017 | 0.255 ± 0.021 | 0.312 ± 0.031 | **0.246** ± 0.023 | **0.274** ± 0.017 | 0.274 ± 0.029 | 0.320 ± 0.041 |

**Main Results.** To answer RQ1, we conduct our proposal on backbone models across seven datasets, and report the overall forecasting accuracy in Table 1. We set the forecasting lengths as $H \in \{96, 192, 336, 720\}$, with the input sequence length $L = 96$. Here, we present the averaged MSE and MAE over four forecasting lengths. Applying TIFO consistently improved the performance of the backbone models across all datasets. More importantly, in datasets with complex frequency characteristics, such as ETTm2, TIFO improves PatchTST and iTransformer by 33.3% ($0.420 \rightarrow 0.280$) and 55.3% ($0.633 \rightarrow 0.283$), respectively. This improvement is attributed to the learned Fourier basis coefficients $\lambda$, allowing these backbones to forecast based on a stationary representation of the input time series.

**Comparison with Baseline Non-stationary Methods.** Table 2 further presents the average comparison results between TIFO and the baseline non-stationary methods, i.e., RevIN, SAN, and FAN. We use the same parameters and forecasting length as in Table 1. For iTransformer, the input sequence length is $L = 96$, and $L = 336$ for DLinear. As shown, TIFO achieves 18 top-1 results and 6 top-2 results out of 28 settings. For instance, in the ETTh1 dataset, TIFO improves the MSE values for DLinear and iTransformer to 0.407 and 0.445, outperforming RevIN (0.460 and 0.463) and SAN (0.421 and 0.466). Similarly, in the Traffic dataset, TIFO improves the MSE value to 0.430, compared to RevIN (0.624), SAN (0.440) and FAN(0.541). Here, TIFO * represents the incorporation of SAN into the backbones, which further improves the second-best results (underlined in the table) to the best.

**Frequency Domain Shift Analysis.** To answer RQ2, we further measure the frequency-domain distribution difference between the train and test dataset amplitude spectra to link accuracy gains to reduced spectral distributional shift. For each frequency $\omega_j$, we gather its amplitudes across all training and testing samples to build two empirical distributions. Jensen–Shannon divergence squared ($\text{JSD}^2$) is a symmetric, bounded average of the forward and reverse Kullback–Leibler divergences, so it can tell us how much the two distributions differ overall (Mateos et al., 2017; Iqbal et al., 2021). Kolmogorov–Smirnov statistic (KS) measures the largest gap between the cumulative distribution functions, highlighting the most significant mismatch between training and testing data (Wang & Wang, 2010). Here, $\text{JSD}^2$ evaluates the overall shift in amplitude distributions, while KS measures the worst-case deviation between training and test data for each frequency component.

Table 3 presents the $\text{JSD}^2$ and KS statistics for the original data (Before) and after applying RevIN, FAN, SAN, and TIFO across four benchmark datasets. Lower values indicate smaller distributional discrepancies between training and test sets. On ETTh1, TIFO reduces $\text{JSD}^2$ from 0.3637 to 0.0435 (a reduction of 88%), and on Electricity from 0.1443 to 0.0423 (71%). KS values also decrease significantly, ranging from 43% to 80% reductions across the same datasets. Figure 5 shows the per-frequency $\text{JSD}^2$ values on the Electricity dataset. Each spoke represents a frequency component, and

Table 2: Multivariate forecasting results (average) with $H \in \{96, 192, 336, 720\}$ for all datasets and fixed input sequence length $L = 96$. The **best** and second best results are highlighted. TIFO * represents the results where both TIFO and SAN are used in the backbones.

| Models | MLP-based (DLinear(Zeng et al., 2023)) | | | | | | | | | | Transformer-based (iTransformer(Liu et al., 2024b)) | | | | | | | | | |
|---|---|---|---|---|---|---|---|---|---|---|---|---|---|---|---|---|---|---|---|---|
| Methods | + TIFO * | | + TIFO | | +SAN | | + FAN | | + RevIN | | + TIFO * | | + TIFO | | +SAN | | + FAN | | + RevIN | |
| Metric | MSE | MAE | MSE | MAE | MSE | MAE | MSE | MAE | MSE | MAE | MSE | MAE | MSE | MAE | MSE | MAE | MSE | MAE | MSE | MAE |
| ETTh1 | 0.413 | 0.424 | **0.407** | **0.419** | 0.421 | 0.427 | 0.469 | 0.473 | 0.460 | 0.456 | 0.455 | 0.449 | **0.445** | **0.443** | 0.466 | 0.455 | 0.504 | 0.488 | 0.463 | 0.452 |
| ETTh2 | 0.339 | **0.384** | 0.337 | **0.384** | 0.342 | 0.387 | 0.459 | 0.469 | 0.561 | 0.518 | 0.378 | 0.408 | **0.376** | **0.400** | 0.392 | 0.413 | 0.543 | 0.515 | 0.385 | 0.412 |
| ETTm1 | **0.341** | **0.372** | 0.357 | 0.375 | 0.344 | 0.376 | 0.367 | 0.392 | 0.413 | 0.407 | **0.389** | **0.398** | 0.396 | 0.406 | 0.401 | 0.406 | 0.424 | 0.428 | 0.406 | 0.410 |
| ETTm2 | **0.255** | 0.316 | 0.256 | **0.313** | 0.260 | 0.318 | 0.278 | 0.339 | 0.350 | 0.413 | 0.285 | 0.334 | **0.283** | **0.327** | 0.287 | 0.336 | 0.337 | 0.385 | 0.294 | 0.337 |
| Electricity | **0.161** | **0.257** | 0.168 | 0.262 | 0.163 | 0.260 | 0.176 | 0.277 | 0.225 | 0.316 | 0.175 | 0.273 | **0.169** | **0.262** | 0.195 | 0.283 | 0.180 | 0.276 | 0.205 | 0.272 |
| Traffic | 0.432 | 0.297 | **0.430** | **0.291** | 0.440 | 0.302 | 0.541 | 0.346 | 0.624 | 0.383 | 0.459 | 0.313 | **0.424** | **0.282** | 0.520 | 0.341 | 0.536 | 0.339 | 0.430 | 0.312 |
| Weather | **0.224** | **0.271** | 0.237 | 0.272 | 0.227 | 0.276 | 0.258 | 0.305 | 0.265 | 0.317 | **0.244** | 0.282 | 0.246 | **0.274** | 0.247 | 0.291 | 0.251 | 0.298 | 0.263 | 0.288 |
| Count | **4** | **4** | 3 | **4** | 0 | 0 | 0 | 0 | 0 | 0 | 2 | 1 | **5** | **6** | 0 | 0 | 0 | 0 | 0 | 0 |

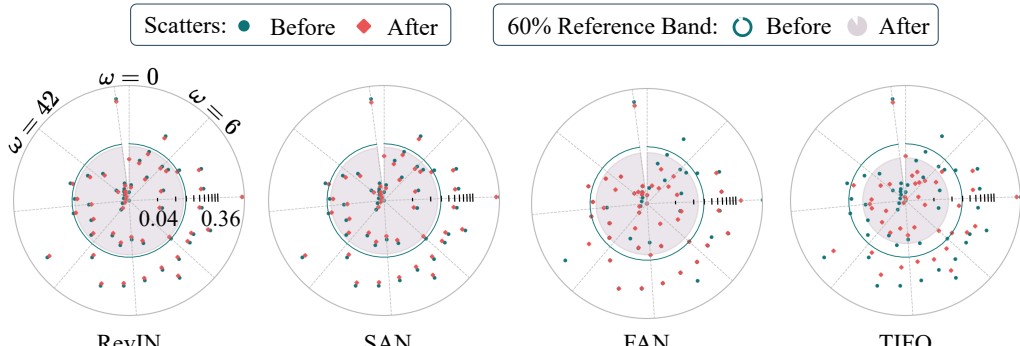

Figure 2: Train-Test Distance Compactness: This figure shows a visualization of the $\text{JSD}^2$ amplitudes distribution distance between the train and test datasets on the electricity data. Each scatter point represents one frequency component. A smaller radius indicates a smaller distributional gap. Green and red colors represent the results before and after applying the learning method, respectively.

Table 3: Frequency-domain distribution distance between the train and test set. Both the $\text{JSD}^2$ ($\downarrow$) and the KS statistic ($\downarrow$) are computed on amplitudes; bold marks the best per row.

| Dataset | Before | | ReVIN | | FAN | | SAN | | TIFO | |
|---|---|---|---|---|---|---|---|---|---|---|
| | $\text{JSD}^2$ | KS | $\text{JSD}^2$ | KS | $\text{JSD}^2$ | KS | $\text{JSD}^2$ | KS | $\text{JSD}^2$ | KS |
| ETTh1 | 0.36367 | 0.35982 | 0.08100 | 0.08937 | 0.14759 | 0.27057 | 0.04745 | 0.08389 | **0.04353** | **0.07357** |
| weather | 0.11156 | 0.19926 | 0.02175 | 0.09172 | 0.05328 | 0.12149 | 0.01739 | 0.09566 | **0.01687** | **0.07934** |
| ECL | 0.14431 | 0.16804 | 0.08060 | 0.11924 | 0.10394 | 0.16537 | 0.07740 | 0.11639 | **0.04225** | **0.09581** |

the gray shadowed circle area serves as a reference band that indicates the region within which $60\%$ of the frequency components fall. A smaller area reflects a lower overall distributional discrepancy. As shown in the figure, TIFO considers all frequency components and significantly reduces distributional differences, achieving effective alignment between training and test datasets. RevIN and SAN, which operate in the time domain, exhibit minimal changes before and after learning, as evidenced by the near overlap between the green line and the gray reference circle. FAN, which explicitly operates in the frequency domain, shows improved performance. However, it focuses only on the top-$k$ frequency components, affecting only a subset of spokes. Overall, TIFO consistently outperforms all baselines, and more results can be found in Appendix C.2.

**Fourier Basis Learning Evaluation** To answer RQ3, after analyzing the impact of TIFO on the distributional difference, we further investigate how TIFO affects the deep forecasting models to learn frequency features. Figure 10 shows the Fourier basis functions in three cases, from left to right are: (i) basis functions before any processing (Before), (ii) after applying FAN, and (iii) after applying TIFO, respectively. We use DLinear as the forecasting model in this evaluation, and results for other models are given in Appendix C.2. The coefficients $\lambda$ of the Fourier basis functions $\zeta_{\omega_{1:4}}$ are $\lambda = 1.0$ in the unprocessed case (Before). FAN sets the $\lambda$ of the top-$k$ high-amplitude frequencies to zero and leaves the $\lambda$ of the rest frequencies unchanged. In contrast, TIFO learns a data-driven $\lambda$

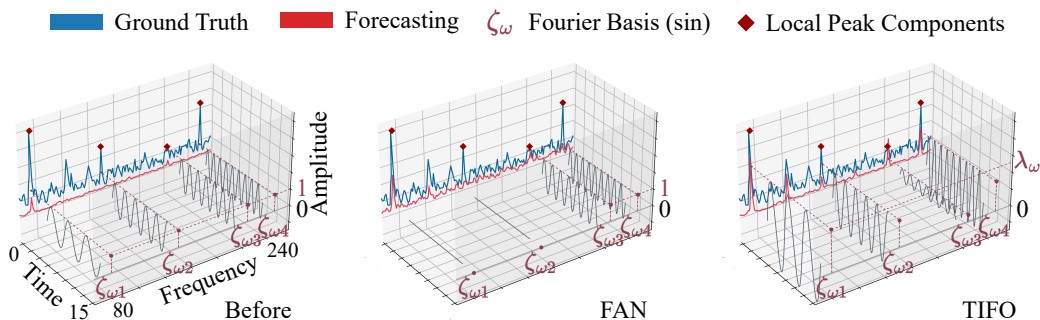

Figure 3: Frequency-domain analysis of Fourier basis Learning. From left to right: unprocessed spectra (Before), after FAN, and after applying TIFO. Each panel is a 3D box with time, frequency, and amplitude axes. We visualize four Fourier basis waves, $\zeta_{\omega_{1:4}}$, to illustrate how each processing method alters the basis functions. In the frequency–amplitude plane, we plot three forecasting cases: the ground truth in blue and the forecasting results based on the processed input in red. The red diamonds mark key local peak frequencies.

Table 4: Running time (s), fixed $L = 96$. Speed-ups over +SAN in parentheses.

| Model | Norm | $H$ | ETTh1 | ETTm1 |
|---|---|---|---|---|
| DLinear | +TIFO | 96 | **3.004** (↑65.5%) | **13.456** (↑63.1%) |
| | | 720 | **3.082** (↑60.5%) | **13.124** (↑64.2%) |
| | +SAN | 96 | 8.688 | 36.706 |
| | | 720 | 8.054 | 36.564 |
| PatchTST | +TIFO | 96 | **7.952** (↑51.6%) | **54.526** (↑14.3%) |
| | | 720 | **8.215** (↑46.2%) | **28.944** (↑56.0%) |
| | +SAN | 96 | 16.226 | 63.686 |
| | | 720 | 15.309 | 65.839 |

Table 5: Ablation study (MAE ↓, mean±std). Bold = best.

| Model | Variant | $H$ | ETTh1 | ETTm1 |
|---|---|---|---|---|
| DLinear | +TIFO | 96 | **0.371**±**0.032** | **0.299**±**0.021** |
| | | 720 | **0.428**±**0.039** | **0.425**±**0.032** |
| | +Random | 96 | 0.379±0.025 | 0.305±0.022 |
| | | 720 | 0.435±0.036 | 0.431±0.031 |
| iTransformer | +TIFO | 96 | **0.389**±**0.023** | **0.330**±**0.035** |
| | | 720 | **0.496**±**0.034** | **0.475**±**0.043** |
| | +Random | 96 | 0.401±0.017 | 0.335±0.033 |
| | | 720 | 0.502±0.029 | 0.483±0.047 |

for each basis function. The line plots display ground truth and corresponding forecasting results in the frequency domain for each case, with local peak components marked by red diamonds. Without any processing, the original model failed to capture these local peaks. FAN, which affects deep forecasting models via frequency modeling, shows improved performance. However, it still failed to capture most local peak components. In contrast, after applying the TIFO, the model successfully captures all four local peak components, outperforms other cases.

**Running Time and Ablation Study.** Table 4 presents the running time results for TIFO and SAN across two datasets. The results show the average time (in seconds per epoch) using DLinear and PatchTST as backbones. TIFO consistently outperforms SAN across all datasets. The full results are in C.3. Notably, we achieved improvements of 60% to 70% in 16 out of 28 experiment settings. These improvements are primarily because TIFO only utilizes FFT and MLP during the training phase, minimizing its impact on computation time. Table 5 shows an ablation in which we replace the computed starting point $s$ with a random vector, while keeping the MLPs and all parameters unchanged. This variant assigns random starting points for the MLPs. The drop in forecasting accuracy compared to TIFO demonstrates that initializing the MLPs with $s$ is essential to help deep forecasting models learn the frequency representations.

**Frequency modeling ablation studies.** To verify the effectiveness of our frequency modeling design, we conduct ablation studies on window functions, frequency resolution, and reconstruction strategies, as summarized in Table 6. First, we analyze the sensitivity to window functions and frequency bins $K$ under the original DFT setting (Table 6a). The results demonstrate that utilizing a basic rectangular window with the full spectrum ($K = 96$) yields the best performance (MSE = 0.3938) with minimal variance. Applying a Hanning window or reducing the resolution ($K = 48$) leads to slight degradation, indicating that preserving the raw global spectral information is preferable.

Table 6: Ablation study on frequency domain modeling on ETTh1 with iTransformer. **(a)** Impact of window functions and resolution under the DFT setting. **(b)** Comparison of reconstruction strategies: DFT vs. Short-Time Fourier Transform (STFT). The default settings with a rectangular window achieves the best performance.



(a) Window & Resolution Settings

| Settings | MSE | MAE |
|---|---|---|
| Rect. ($K = 48$) | $0.4107 \pm 0.0125$ | $0.4270 \pm 0.0130$ |
| **Ori (Rect., $K = 96$)** | $\mathbf{0.3938 \pm 0.0036}$ | $\mathbf{0.4104 \pm 0.0036}$ |
| Hann ($K = 48$) | $0.4117 \pm 0.0015$ | $0.4250 \pm 0.0010$ |
| Hann ($K = 96$) | $0.4110 \pm 0.0020$ | $0.4243 \pm 0.0015$ |

(b) Frequency Decomposition Strategy

| Strategy | MSE | MAE |
|---|---|---|
| **Ori (DFT, Rect.)** | $\mathbf{0.3938 \pm 0.0036}$ | $\mathbf{0.4104 \pm 0.0036}$ |
| STFT (Hann, 50% ovlp) | $0.5820 \pm 0.0174$ | $0.5118 \pm 0.0056$ |
| STFT (Hann, 75% ovlp) | $0.5814 \pm 0.0166$ | $0.5110 \pm 0.0052$ |
| STFT ($\sqrt{\text{Hann}}$, 50% ovlp) | $0.4005 \pm 0.0024$ | $0.4148 \pm 0.0024$ |



Second, we compare original TIFO against Short-Time Fourier Transform (STFT) variants (Table 6b). Standard STFT configurations with Hanning windows and overlap (50% or 75%) significantly impair accuracy (MSE $> 0.58$), likely due to boundary artifacts introduced by windowing. The STFT variant using a $\sqrt{\text{Hann}}$ window fails to outperform the original TIFO (MSE $= 0.4005$ vs. $0.3938$).

## 5 RELATED WORKS

**Time-domain normalization.** To mitigate distribution shifts, existing methods aim to align distributions by normalizing statistics of each sample. RevIN (Kim et al., 2021) performs instance-wise z-score normalization, while Dish-TS (Fan et al., 2023) utilizes learnable mean and variance parameters. Further improvements include SAN (Liu et al., 2023), which models non-stationarity in fine-grained sub-series, and SIN (Han et al., 2024), which employs an adaptive network to learn normalization objectives. These methods do not explicitly observe the overall energy distribution of the training set in the frequency domain. In contrast, TIFO introduces a dataset-level stationarity weighting module. By calculating the cross-sample stability of each frequency component across the entire training set, TIFO explicitly targets the spectral misalignment that remains unaddressed by methods like RevIN and SAN.

**Frequency selection and masking.** Recognizing the importance of spectral features, recent works have explored selecting informative frequency components. For instance, FAN (Ye et al., 2024) takes an initial step by learning predominant frequencies as non-stationarities, employing a heuristic top-$k$ masking strategy to zero out a subset of components based on amplitude. Similarly, methods like Zhou et al. (2022b); Woo et al. (2022); Zhou et al. (2022a) utilize sparse attention mechanisms to select frequencies. However, these approaches typically rely on attention mechanism or top-$k$ selection derived from the instantaneous high-energy frequencies of a single sample or local window. This heuristic selection risks limiting the model to accidentally high-energy frequencies that are unstable at the dataset level, while inadvertently discarding stable but low-energy periodic patterns. Unlike these methods, TIFO learns a set of continuous weights for Fourier basis functions based on dataset-level stationarity. This allows TIFO to suppress non-stationary frequencies while preserving stationary signals, preventing the over-suppression or mis-selection common in heuristic approaches.

**Frequency domain models.** Frequency domain modeling has proven highly effective for capturing global dependencies and improving efficiency. Mainstream architectures, such as Autoformer (Wu et al., 2021), Fedformer (Zhou et al., 2022b), TimesNet (Wu et al., 2023), and Frequency MLP (Yi et al., 2023), move primary operators (e.g., attention or convolution) into the frequency domain. Recent works have further explored spectral properties: FITS (Xu et al., 2024) utilizes frequency interpolation and low-pass filtering for parameter-efficient modeling; FreDF (Wang et al., 2025) introduces a frequency-domain loss function to debias direct forecasting against label autocorrelation; and Dynamic Fusion (Zhang et al., 2024a) optimizes the combination of independent frequency predictions. However, these methods typically focus on optimizing architecture efficiency (FITS), output-side loss constraints (FreDF), or representation expressiveness (Dynamic Fusion). They do not explicitly estimate dataset-level frequency statioanrity to guide input adaptation. In contrast, TIFO is designed as a universal preprocessing layer decoupled from the backbone architecture. It focuses specifically on learning frequency stationarity from training statistics to mitigate input-side spectral distribution shift. Therefore, TIFO is complementary to, rather than repetitive of, these architecture-level frequency domain models.

# 6 REPRODUCIBILITY STATEMENT

To complement the model presented in Section 3.1, we provide pseudocode and an anonymous repository that illustrate our approach. Algorithm 1 shows how frequency-wise stability $s$ is computed from the training data. Algorithm 2 demonstrates how these scores are used to adaptively re-weight Fourier coefficients through independent MLP mappings, before transforming back to the time domain. The complete implementation, including model training and experimental setup, is available at our anonymous repository: https://anonymous.4open.science/r/TIFO-6BE1.

| **Algorithm 1:** Compute Stability Scores $s_j$ | **Algorithm 2:** Apply Fourier Coefficients $\lambda_j$ |
|---|---|

```python
def compute_stability_scores(X):
    """
    X: (N, T) training set
    """
    amp = np.abs(np.fft.fft(X, axis=1))
    mean = amp.mean(axis=0)
# per-freq mean
    std = amp.std(axis=0) + 1e-5
# per-freq std
    return mean / std # s_j
```

```python
def weight_spectra(x, s, f_cos, f_sin):
    """
    f_*: learnable MLP layers
    """
    X = np.fft.fft(x)
    X[0::2] *= f_cos(s[0::2])
# re-weight real parts
    X[1::2] *= f_sin(s[1::2])
# re-weight imag parts
    return np.fft.ifft(X, n=len(x))
```

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

Appendix / Supplemental Material

**Contents**

## A    Conclusion

Nonstationary time series forecasting suffers from distributional shift due to the different distributions that produce the training and test data. As a result, a model trained on the training data may perform poorly on the test data. These distributions can be regarded as governed by a time structure. A time series can be considered as first sampling from a time structure distribution, then from a time-conditional observation distribution. Existing methods attempt to alleviate the issue by normalizing the distributions. To this end, we propose a Time-Invariant Frequency Operator (TIFO), which learns stationarity-aware weights over the frequency spectrum across the entire dataset. The weight representation highlights stationary frequency components while suppressing non-stationary ones, thereby mitigating the distribution shift issue in time series. Extensive experiments demonstrate that the proposed method achieves superior performance, yielding 18 top-1 and 6 top-2 results out of 28 settings compared to the baselines.

## B    Additional Discussions and Details

### B.1    Additional Discussion about background

Deep learning models have demonstrated significant success in time series forecasting (Moosavi et al., 2019; Zhou et al., 2021; Wu et al., 2021; M. et al., 2022; Nie et al., 2023; Zhang & Yan, 2023; Liu et al., 2024b). These models aim to extract diverse and informative patterns from historical observations to enhance the accuracy of future time series predictions. To achieve accurate time

series predictions, a key challenge is that time series data derived from numerous real-world systems exhibit dynamic and evolving patterns, i.e., a phenomenon known as non-stationarity (Stoica et al., 2005; Box et al., 2015; Xie et al., 2018; Rhif et al., 2019).

This characteristic typically results in discrepancies among training, testing, and future unseen data distributions. Consequently, the non-stationary characteristics of time series data necessitate the development of forecasting models that are robust to such temporal shifts in data distribution, while failing to address this challenge often leads to representation degradation and compromised model generalization (Kim et al., 2021; Du et al., 2021).

To provide more research background details, we conduct a schematic case study using a synthetic time series dataset. As shown in Figure 4, for each temporal condition $t_i$, we generate 50 independent samples, each governed by different temporal structures. Here, we generate different temporal structures by mixing low- and high-frequency components, as shown in the Figure 4 (bottom). For example, $t_1$ mainly contains low-frequency features, while the energy distribution of $t_4$ primarily exists in the high-frequency bands.

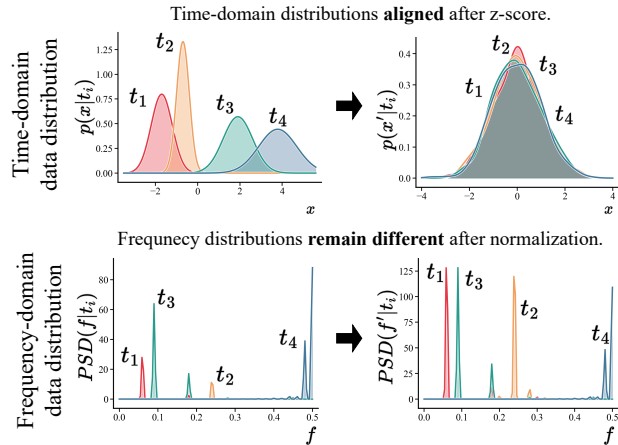

Figure 4: Illustration of z-score normalization in both time and frequency domains. (Top) Data generating distributions $p(x|t_i)$ across different temporal structures $t_i$ are aligned after $z$-score, sharing a common location and scale. (Bottom) Frequency-domain power spectra before and after $z$-score. The frequency-domain distribution remains divergent after normalization.

Then, we use the numerical distribution of $x$ and the Power Spectral Density (PSD) to evaluate the data distribution in both the time domain and the frequency domain, respectively. The figure visualizes the data before and after applying z-score normalization. In the time domain, normalization successfully transfers the distributions from different $t_i$ into a common Gaussian shape, thereby reducing distributional gaps. However, when examining the same data in the frequency domain, we find a very different result: the spectral energy distributions remain distinct and non-overlapping, revealing that the z-score does not address discrepancies in temporal dependencies or frequency compositions.

This is because the z-score operates only on first- and second-order statistics, ignoring higher-order temporal structure information. In contrast, periodicity and other temporal dynamics are preserved in the signal and are clearly reflected in the frequency features. In other words, while z-score normalization alters the numerical distribution of the data, it preserves the underlying temporal organization. Thus, the distribution shift in the frequency domain remains unchanged.

## B.2 RELATED WORKS

**Time Series Forecasting.** Transformer-based architectures have become the mainstream in time series forecasting (Nie et al., 2023; Zhang & Yan, 2023; Jiang et al., 2023; Liu et al., 2024b). Meanwhile, simple multilayer perceptron (MLP) models, such as DLinear (Zeng et al., 2023; Chen et al., 2023; Zhou et al., 2022a), have also attracted attention due to their lower computational costs and forecasting accuracy comparable to transformer-based models.

**Normalization-based Methods.** Existing methods aim to quantify and explicitly eliminate non-stationary components from both training and test data, thereby aligning distributions and enhancing generalization. A normalization is performed by subtracting the empirical mean and dividing by the variance computed from the data. At the forecasting stage, denormalization is applied to reintroduce

these descriptive statistics to model outputs. RevIN (Kim et al., 2021) is an innovative work that focuses on z-score normalization. Dish-TS (Fan et al., 2023) utilizes learned mean and variance for denormalization. SAN (Liu et al., 2023) models non-stationarity in a set of fine-grained sub-series and proposes an additional loss function to predict their statistics. Instead of mean and variance, SIN (Han et al., 2024) proposes an independent neural network to learn features as the objectives of normalization and denormalization adaptively. However, existing methods focus on modeling statistical variations in the time domain. A recent FAN (Ye et al., 2024) has taken an initial step in learning predominant frequency components as non-stationarities. FAN uses heuristic top-$k$ masking of the largest-amplitude frequencies, zeroing out a subset of components. However, this heuristic selection may risk discarding critical periodic patterns embedded in high-energy regions and introduce sub-optimal frequency correlations into the model, inadvertently misleading the model training.

**Frequency Domain Modeling.** Frequency domain modeling has proven highly effective in time series forecasting. Mainstream works employ neural networks to automatically learn frequency representations directly in the raw Fourier domain (Wu et al., 2021; Zhou et al., 2022b; Wang et al., 2022; Wu et al., 2023; Yi et al., 2023), but such approaches can be vulnerable to noise and to frequency components that vary significantly over time. Some other methods are designed to select informative components via sparse selection (Zhou et al., 2022b; Woo et al., 2022; Zhou et al., 2022a; Ye et al., 2024) or local normalization in the frequency domain (Piao et al., 2024), yet these still rely on heuristics, such as top-$k$ selection, or rely on the model to identify the key frequency features. In contrast, our method learns and adjusts the coefficients of the Fourier basis functions, naturally encoding the relative importance of different spectral components and allowing seamless deployment on any forecasting backbone. Notably, FAN uses heuristic top-$k$ masking of the largest-amplitude frequencies, zeroing out a subset of components. In contrast, TIFO learns a data-driven coefficient for every Fourier basis function, enabling dynamic weighting of the contribution of each component.

### B.3 ABLATION STUDY ON STATIONARITY METRICS

To validate the effectiveness of our proposed stability metric $S(k,c) = \mu_{k,c}/\sigma_{k,c}$ (the reciprocal of the Coefficient of Variation), we conducted an ablation study comparing it against two alternative metrics: Spectral Entropy and Correlation (a proxy for Mutual Information).

**Rationale for Metric Selection.** The core hypothesis of TIFO is that frequency components exhibiting stable behavior across different samples in the training set are more reliable for generalization.

- **Coefficient of Variation (CV, ours):** This statistic explicitly measures the dispersion of a variable across multiple observations relative to its mean. In our context, $\mu/\sigma$ quantifies the *cross-sample stationarity*. A high score indicates that a frequency component consistently appears with stable energy across the entire dataset, aligning perfectly with our objective.

- **Spectral Entropy:** This metric measures the complexity or "spikiness" of the power spectrum within a *single* signal window. While useful for characterizing *intra-sample* complexity, it does not capture how a specific frequency evolves or drifts across the dataset (distribution shift).

- **Mutual Information (MI):** MI measures the dependency between the input frequencies and the forecasting target. While theoretically strong, calculating MI typically requires access to future labels (forecasting targets), making it a supervised signal. Our goal for TIFO is to serve as an unsupervised, plug-and-play regularization layer that relies solely on input statistics.

**Experimental Results.** We compared these metrics on the ETTh1 dataset using iTransformer as the backbone. The training was set to a maximum of 30 epochs with early stopping (patience=5).

- `mu_sigma` (Ours): The default $\mu/\sigma$ metric.
- `entropy`: Inverse spectral entropy (lower entropy → higher weight).
- `corr`: The Pearson correlation coefficient $|r|$ between the frequency amplitude and the mean of the future target series, serving as a proxy for Mutual Information.

The results are summarized in Table 7. Our proposed `mu_sigma` achieves the best performance in both MSE and MAE. Notably, it outperforms `entropy`, suggesting that consistency across the

dataset (cross-sample stationarity) is more critical for handling distribution shift than intra-sample complexity. Furthermore, `mu_sigma` performs comparably to (or slightly better than) the target-aware `corr` metric. This confirms that our unsupervised stability score is highly effective and sufficient for identifying robust frequency patterns without requiring label guidance or complex supervised computation.

Table 7: Comparison of different stationarity metrics on ETTh1 with iTransformer. *mu_sigma* denotes our default unsupervised metric; *entropy* focuses on intra-sample distribution; *corr* utilizes future labels to approximate Mutual Information.

| Metric | Description | MSE | MAE |
|---|---|---|---|
| mu_sigma (**Ours**) | Mean/Std (Unsupervised, Cross-sample stability) | **0.3904** | **0.4071** |
| entropy | Spectral Entropy (Intra-sample complexity) | 0.3962 | 0.4106 |
| corr | Correlation $|r|$ (Target-aware, MI approx.) | 0.3948 | 0.4077 |

### B.4 DETAILS OF THE DATASETS.

Weather contains 21 channels (e.g., temperature and humidity) and is recorded every 10 minutes in 2020. ETT (Zhou et al., 2021) (Electricity Transformer Temperature) consists of two hourly-level datasets (ETTh1, ETTh2) and two 15-minute-level datasets (ETTm1, ETTm2). Electricity (Lai et al., 2018), from the UCI Machine Learning Repository and preprocessed by, is composed of the hourly electricity consumption of 321 clients in kWh from 2012 to 2014. Solar-Energy (Lai et al., 2018) records the solar power production of 137 PV plants in 2006, sampled every 10 minutes. Traffic contains hourly road occupancy rates measured by 862 sensors on San Francisco Bay area freeways from January 2015 to December 2016. More details of these datasets can be found in Table.8.

Table 8: Overview of Datasets

| Dataset | Source | Resolution | Channels | Time Range |
|---|---|---|---|---|
| Weather | Autoformer(Wu et al., 2021) | Every 10 minutes | 21 (e.g., temperature, humidity) | 2020 |
| ETTh1 | Informer(Zhou et al., 2021) | Hourly | 7 states of a electrical transformer | 2016-2017 |
| ETTh2 | Informer(Zhou et al., 2021) | Hourly | 7 states of a electrical transformer | 2017-2018 |
| ETTm1 | Informer(Zhou et al., 2021) | Every 15 minutes | 7 states of a electrical transformer | 2016-2017 |
| ETTm2 | Informer(Zhou et al., 2021) | Every 15 minutes | 7 states of a electrical transformer | 2017-2018 |
| Electricity | UCI ML Repository | Hourly | 321 clients' consumption | 2012-2014 |
| Traffic | Informer(Zhou et al., 2021) | Hourly | 862 sensors' occupancy | 2015-2016 |

### B.5 PREPROCESSING AND EVALUATION DETAILS

Given the raw multivariate time-series data $X \in \mathbb{R}^{T \times C}$, we first slide a window of length $L = 96$ over $X$ with stride 1. This produces overlapping segments $W = \{ X_{i:i+L-1} \mid i = 1, \ldots, T - L + 1 \}$. Each segment is of shape $(L, C)$. We order the segments chronologically and split $W$ into training (70%), validation (20%), and test (10%) sets. On the training set, we compute per-variable means $\mu_j$ and standard deviations $\sigma_j$ for $j = 1, \ldots, C$. We then apply per-variable $z$-score normalization to every segment in all splits: $\tilde{w}_{i,j} = \frac{w_{i,j} - \mu_j}{\sigma_j}, \quad \forall w \in W, \ i = \{1, \ldots, L\}, \ j = \{1, \ldots, C\}$. The normalized segments form $\mathcal{D}_{\text{train}}$, $\mathcal{D}_{\text{val}}$, and $\mathcal{D}_{\text{test}}$, ready for model training and evaluation.

---

**Algorithm 3:** Pre-processing for Time-Series Forecasting

---

**Input:** Time-series $X \in \mathbb{R}^{T \times C}$, window length $L$, split ratios $(0.7, 0.2, 0.1)$
**Output:** Normalized sets $\mathcal{D}_{\text{train}}, \mathcal{D}_{\text{val}}, \mathcal{D}_{\text{test}}$
Slice $X$ into overlapping windows $W \leftarrow \{X_{i:i+L-1} \mid i = 1, \ldots, T - L + 1\}$;
Split $W$ into train/val/test by first 70%, next 20 %, last 10 %;
Compute $\mu_j = \text{mean}(\{w_{i,j} \mid w \in \mathcal{D}_{\text{train}}\})$;
Compute $\sigma_j = \text{std}(\{w_{i,j} \mid w \in \mathcal{D}_{\text{train}}\})$;
**foreach** *segment* $w \in W$ **do**
    **for** *variable* $j = 1, \ldots, C$ **do**
        $\tilde{w}_j \leftarrow (w_j - \mu_j)/\sigma_j$;
    **end**
    Add $\tilde{w}$ to its split's dataset;
**end**
**return** $\mathcal{D}_{\text{train}}, \mathcal{D}_{\text{val}}, \mathcal{D}_{\text{test}}$

---

### B.6 DETAILS OF THE BASELINES

**Reversible Instance Normalization.** Reversible Instance Normalization (Revin) normalizes each input sample using z-score normalization while preserving the original mean and variance. Revin reverses the normalization to model outputs by using the saved statistics and applies learnable scaling and shifting parameters ($\gamma$ and $\beta$).

**Sequential Adaptive Normalization.** Sequential Adaptive Normalization (SAN) has two training phases. In the first phase, SAN is trained to learn the relationships between patches of input and target data by mapping their means and variances. In the second phase, SAN parameters are frozen, and only the forecasting model is trained. During inference, input data is normalized using SAN, and the model output is reverse-normalized with predicted statistics by SAN.

**Frequency Adaptive Normalization (FAN)** is a deep learning approach for time series forecasting that decomposes input sequences into frequency components using FFT/RFFT. The algorithm separates the top-k dominant frequency components from residual signals, then employs an MLPfreq network to model the main frequency patterns while handling residuals separately. By processing frequency and temporal information through parallel pathways and combining them via learnable weights, FAN achieves improved forecasting accuracy through frequency-domain feature extraction and adaptive normalization. The method is particularly effective for capturing periodic patterns and long-term dependencies in time series data.

---

**Algorithm 4:** Reversible Instance Normalization (Revin)

---

**Input:** Time-series data $X$, Forecasting model $\mathcal{F}$
**Output:** Forecasted data $\hat{X}$ ;
**for** *each instance $X_i$ in $X$* **do**
    Compute mean $\mu_i \leftarrow \text{mean}(X_i)$;
    Compute variance $\sigma_i^2 \leftarrow \text{variance}(X_i)$ ;
    Normalize $\tilde{X}_i \leftarrow \frac{X_i - \mu_i}{\sigma_i}$;
    **Store** $\mu_i$ and $\sigma_i^2$ ;
**end**
$\tilde{X} \leftarrow \{\tilde{X}_1, \tilde{X}_2, \ldots, \tilde{X}_N\}$ ;
$\tilde{Y} \leftarrow \mathcal{F}(\tilde{X})$ ;
**for** *each forecasted instance $\tilde{Y}_i$* **do**
    Reverse Normalize $Y_i \leftarrow \tilde{Y}_i \times \sigma_i + \mu_i$ ;
    Apply learnable parameters $Y_i \leftarrow \gamma \times Y_i + \beta$ ;
**end**
**return** $\hat{X} = \{Y_1, Y_2, \ldots, Y_N\}$ ;

---

---

**Algorithm 5:** Sequential Adaptive Normalization (SAN)

---

**Stage 1: Train SAN**;

**Input:** Training data $X$ and targets $Y$ Divide $X$ and $Y$ into patches $\{X_p\}$ and $\{Y_p\}$

**for** *each pair of patches* $(X_p, Y_p)$ **do**

    Compute means $\mu_X \leftarrow \text{mean}(X_p)$, $\mu_Y \leftarrow \text{mean}(Y_p)$ ;

    Compute variances $\sigma_X^2 \leftarrow \text{variance}(X_p)$, $\sigma_Y^2 \leftarrow \text{variance}(Y_p)$ ;

    Train SAN to map $(\mu_X, \sigma_X^2)$ to $(\mu_Y, \sigma_Y^2)$ using loss on $\mu_Y$ and $\sigma_Y^2$ ;

    **Stage 2: Train Forecasting Model** Freeze SAN parameters;

    **for** *each training iteration* **do**

        Divide input $X$ into patches $\{X_p\}$ ;

        **for** *each patch* $X_p$ **do**

            Normalize $X_p \leftarrow \frac{X_p - \mu_X}{\sigma_X}$ using SAN's learned $\mu_X$ and $\sigma_X^2$

        **end**

        Forecast $\tilde{Y} \leftarrow \mathcal{F}(X)$ ;

        Divide $\tilde{Y}$ into patches $\{\tilde{Y}_p\}$ ;

        **for** *each forecasted patch* $\tilde{Y}_p$ **do**

            Predict $\mu_Y, \sigma_Y^2$ using SAN Reverse Normalize $Y_p \leftarrow \tilde{Y}_p \times \sigma_Y + \mu_Y$

        **end**

    **end**

    Compute loss $\mathcal{L}(Y, \hat{Y})$ ;

    Update forecasting model parameters $\theta$ via backpropagation ;

**end**

**return** Trained forecasting model $\mathcal{F}$ ;

---

## B.7 DETAILS OF THE BACKBONES AND SETUP

In our study, we selected three distinct forecasting models to evaluate the effectiveness of our proposed normalization techniques. DLinear is an MLP-based model renowned for its lightweight architecture, utilizing two separate multilayer perceptrons (MLPs) to learn the periodic and trend components of the data independently.

PatchTST and iTransformer are both Transformer-based models with unique approaches to handling time-series data. PatchTST introduces a patching operation that samples each input time series into multiple patches, which are then used as input tokens for the transformer, effectively capturing local temporal patterns. In contrast, iTransformer emphasizes channel-wise attention by treating the entire sequence of each channel as a transformer token and employing self-attention mechanisms to learn the relationships between different channels.

For all models, we first compute the starting point (Coefficient of variation) across the entire training dataset, a fixed computational process that typically takes less than five seconds. Following this, we apply a simple, parameter-free normalization and denormalization method. After normalization, the input data is processed through our custom weighting layer before being fed into the forecasting models.

## B.8 OTHER EXPERIMENTS DETAILS

**Loss Function.** For our experiments, we adhere to a conventional approach by employing the Mean Squared Error (MSE) loss function, implemented as `nn.MSELoss` in our framework. The MSE loss quantifies the average squared difference between the predicted values and the actual target values, providing a straightforward measure of prediction accuracy. Mathematically, the MSE loss is expressed as $\mathcal{L}_{\text{MSE}} = \frac{1}{N} \sum_{i=1}^{N} (\hat{y}_i - y_i)^2$, where $N$ is the number of samples, $\hat{y}_i$ represents the predicted value, and $y_i$ denotes the true target value for the $i$-th sample. This loss function effectively penalizes larger errors more heavily, encouraging the model to achieve higher precision in its predictions.

---

**Algorithm 6:** Frequency Adaptive Normalization (FAN)

---

**Stage 1: Frequency Decomposition**;

**Input:** Input sequence $x \in \mathbb{R}^{B \times T \times N}$, top-k frequency components $k$, RFFT flag

**Function** MAINFREQPART($x, k$, rfft);

**if** *rfft = True* **then**
  | $x_f \leftarrow$ RFFT($x$, dim = 1) ;

**else**
  | $x_f \leftarrow$ FFT($x$, dim = 1) ;

**end**

indices $\leftarrow$ TopK($|x_f|, k$, dim = 1) ;

mask $\leftarrow$ zeros_like($x_f$) ;

mask.scatter(indices, 1) ;

$x_f^{\text{filtered}} \leftarrow x_f \odot$ mask ;

**if** *rfft = True* **then**
  | $x^{\text{filtered}} \leftarrow$ IRFFT($x_f^{\text{filtered}}$, dim = 1) ;

**else**
  | $x^{\text{filtered}} \leftarrow$ IFFT($x_f^{\text{filtered}}$, dim = 1) ;

**end**

$x^{\text{residual}} \leftarrow x - x^{\text{filtered}}$ ;

**return** $x^{\text{residual}}, x^{\text{filtered}}$ ;

**Stage 2: Train FAN Model**;

**Input:** Training data $X$, sequence length $T$, prediction length $O$, channels $N$, freq_topk $k$

Initialize MLPfreq model $\mathcal{M}_{\text{freq}}$ with parameters $\theta_{\text{freq}}$ ;

Initialize learnable weights $w \in \mathbb{R}^{2 \times N}$ ;

**for** *each training iteration* **do**
  **Normalization Phase:**;
  $(x^{\text{residual}}, x^{\text{filtered}}) \leftarrow$ MAINFREQPART($X, k$, rfft) ;
  $\hat{x}^{\text{main}} \leftarrow \mathcal{M}_{\text{freq}}(x^{\text{filtered}}, X)$ ;
  **Forward Pass:**;
  $x^{\text{norm}} \leftarrow x^{\text{residual}}$ ;
  **Denormalization Phase:**;
  $\hat{x}^{\text{residual}} \leftarrow$ ForecasterOutput ;
  $\hat{x}^{\text{final}} \leftarrow \hat{x}^{\text{residual}} + \hat{x}^{\text{main}}$ ;
  **Loss Computation:**;
  $(y^{\text{residual}}, y^{\text{main}}) \leftarrow$ MAINFREQPART($Y_{\text{true}}, k$, rfft) ;
  $\mathcal{L} \leftarrow$ MSE($\hat{x}^{\text{main}}, y^{\text{main}}$) + MSE($\hat{x}^{\text{residual}}, y^{\text{residual}}$) ;
  Update parameters $\theta_{\text{freq}}$ and $w$ via backpropagation ;

**end**

**Stage 3: MLPfreq Architecture**;

**Input:** Main frequency signal $x^{\text{main}} \in \mathbb{R}^{B \times N \times T}$, original input $x \in \mathbb{R}^{B \times N \times T}$

$h_{\text{freq}} \leftarrow$ ReLU(Linear$_{T \to 64}(x^{\text{main}})$) ;

$h_{\text{concat}} \leftarrow$ Concat($[h_{\text{freq}}, x]$, dim = −1) ;

$h_{\text{hidden}} \leftarrow$ ReLU(Linear$_{(64+T) \to 128}(h_{\text{concat}})$) ;

output $\leftarrow$ Linear$_{128 \to O}(h_{\text{hidden}})$ ;

**return** Trained FAN model with frequency decomposition capability ;

---

**Computational Resources.** All experiments were conducted on an NVIDIA RTX A6000 GPU with 48GB of memory, utilizing CUDA version 12.4 for accelerated computation. This high-performance computational setup facilitated efficient training and evaluation of our forecasting models, ensuring timely execution of experiments even with large-scale time-series data.

## C  THE FULL RESULTS.

### C.1  FULL LONG-TERM FORECASTING RESULTS.

Table 9 and Table 10 present the full comprehensive results discussed in the main paper. This table includes the prediction accuracy outcomes on the [Electricity, ETTh1, ETTh2, ETTm1, ETTm2, Traffic, Weather] dataset, utilizing the [DLinear, PatchTST, iTransformer] as the backbone model. We have compared our method against all baseline models across all forecasting horizons ($H \in \{96, 192, 336, 720\}$).

Table 9: Detailed results of comparing our proposal and other normalization methods. The best results are highlighted in **bold**. The second best are underlined.

| | | DLinear (Zeng et al., 2023) + Ours* MSE | MAE | + Ours MSE | MAE | + SAN MSE | MAE | + RevIN MSE | MAE | PatchTST (Nie et al., 2023) + Ours* MSE | MAE | + Ours MSE | MAE | + SAN MSE | MAE | + RevIN MSE | MAE | iTransformer (Liu et al., 2024b) + Ours* MSE | MAE | + Ours MSE | MAE | + SAN MSE | MAE | + RevIN MSE | MAE |
|---|---|---|---|---|---|---|---|---|---|---|---|---|---|---|---|---|---|---|---|---|---|---|---|---|---|
| Electricity | 96 | 0.135 | 0.230 | 0.140 | 0.237 | 0.137 | 0.234 | 0.210 | 0.278 | 0.175 | 0.266 | 0.190 | 0.280 | 0.182 | 0.271 | 0.212 | 0.297 | 0.145 | 0.244 | 0.143 | 0.237 | 0.171 | 0.262 | 0.152 | 0.251 |
| | 192 | 0.149 | 0.245 | 0.155 | 0.249 | 0.151 | 0.247 | 0.210 | 0.304 | 0.183 | 0.273 | 0.195 | 0.286 | 0.186 | 0.276 | 0.213 | 0.300 | 0.169 | 0.266 | 0.159 | 0.252 | 0.180 | 0.270 | 0.165 | 0.255 |
| | 336 | 0.165 | 0.262 | 0.171 | 0.267 | 0.166 | 0.264 | 0.223 | 0.309 | 0.198 | 0.289 | 0.211 | 0.301 | 0.200 | 0.290 | 0.227 | 0.314 | 0.178 | 0.271 | 0.172 | 0.266 | 0.194 | 0.284 | 0.180 | 0.272 |
| | 720 | 0.198 | 0.291 | 0.208 | 0.298 | 0.201 | 0.295 | 0.257 | 0.349 | 0.233 | 0.317 | 0.253 | 0.334 | 0.237 | 0.322 | 0.268 | 0.344 | 0.210 | 0.311 | 0.205 | 0.295 | 0.237 | 0.319 | 0.227 | 0.312 |
| ETTh1 | 96 | 0.375 | 0.398 | 0.371 | 0.392 | 0.383 | 0.399 | 0.396 | 0.410 | 0.380 | 0.401 | 0.374 | 0.395 | 0.387 | 0.405 | 0.392 | 0.413 | 0.380 | 0.400 | 0.389 | 0.404 | 0.398 | 0.411 | 0.394 | 0.409 |
| | 192 | 0.410 | 0.417 | 0.404 | 0.412 | 0.419 | 0.419 | 0.445 | 0.440 | 0.442 | 0.439 | 0.424 | 0.428 | 0.445 | 0.440 | 0.448 | 0.436 | 0.429 | 0.427 | 0.447 | 0.440 | 0.438 | 0.435 | 0.460 | 0.449 |
| | 336 | 0.430 | 0.427 | 0.426 | 0.426 | 0.437 | 0.432 | 0.487 | 0.465 | 0.480 | 0.456 | 0.471 | 0.452 | 0.505 | 0.471 | 0.489 | 0.456 | 0.479 | 0.451 | 0.492 | 0.463 | 0.481 | 0.456 | 0.501 | 0.475 |
| | 720 | 0.437 | 0.455 | 0.428 | 0.448 | 0.446 | 0.459 | 0.512 | 0.510 | 0.519 | 0.501 | 0.514 | 0.500 | 0.527 | 0.507 | 0.525 | 0.503 | 0.491 | 0.471 | 0.496 | 0.482 | 0.528 | 0.502 | 0.521 | 0.504 |
| ETTh2 | 96 | 0.273 | 0.335 | 0.273 | 0.336 | 0.277 | 0.338 | 0.344 | 0.397 | 0.292 | 0.347 | 0.301 | 0.349 | 0.314 | 0.361 | 0.344 | 0.397 | 0.298 | 0.352 | 0.297 | 0.345 | 0.302 | 0.354 | 0.300 | 0.349 |
| | 192 | 0.335 | 0.374 | 0.336 | 0.376 | 0.340 | 0.378 | 0.485 | 0.481 | 0.385 | 0.402 | 0.380 | 0.399 | 0.391 | 0.421 | 0.389 | 0.411 | 0.371 | 0.402 | 0.380 | 0.395 | 0.383 | 0.402 | 0.381 | 0.415 |
| | 336 | 0.361 | 0.399 | 0.355 | 0.395 | 0.356 | 0.398 | 0.582 | 0.536 | 0.431 | 0.438 | 0.410 | 0.424 | 0.444 | 0.466 | 0.437 | 0.451 | 0.425 | 0.435 | 0.420 | 0.428 | 0.435 | 0.441 | 0.433 | 0.442 |
| | 720 | 0.388 | 0.429 | 0.384 | 0.423 | 0.396 | 0.435 | 0.836 | 0.659 | 0.429 | 0.461 | 0.422 | 0.443 | 0.467 | 0.484 | 0.430 | 0.481 | 0.420 | 0.444 | 0.410 | 0.432 | 0.448 | 0.457 | 0.426 | 0.445 |
| ETTm1 | 96 | 0.285 | 0.339 | 0.299 | 0.341 | 0.288 | 0.342 | 0.353 | 0.374 | 0.322 | 0.359 | 0.321 | 0.362 | 0.325 | 0.361 | 0.353 | 0.374 | 0.326 | 0.361 | 0.330 | 0.370 | 0.331 | 0.373 | 0.341 | 0.376 |
| | 192 | 0.321 | 0.359 | 0.336 | 0.364 | 0.323 | 0.363 | 0.391 | 0.392 | 0.350 | 0.379 | 0.365 | 0.381 | 0.355 | 0.381 | 0.391 | 0.401 | 0.365 | 0.384 | 0.374 | 0.391 | 0.376 | 0.381 | 0.380 | 0.394 |
| | 336 | 0.355 | 0.380 | 0.370 | 0.383 | 0.357 | 0.384 | 0.423 | 0.413 | 0.381 | 0.401 | 0.407 | 0.408 | 0.385 | 0.402 | 0.423 | 0.413 | 0.395 | 0.403 | 0.408 | 0.414 | 0.412 | 0.418 | 0.419 | 0.418 |
| | 720 | 0.405 | 0.411 | 0.425 | 0.414 | 0.409 | 0.415 | 0.486 | 0.449 | 0.446 | 0.436 | 0.464 | 0.442 | 0.450 | 0.437 | 0.486 | 0.459 | 0.471 | 0.447 | 0.475 | 0.449 | 0.485 | 0.453 | 0.486 | 0.455 |
| ETTm2 | 96 | 0.163 | 0.255 | 0.165 | 0.254 | 0.166 | 0.258 | 0.194 | 0.293 | 0.177 | 0.272 | 0.179 | 0.262 | 0.184 | 0.277 | 0.185 | 0.272 | 0.178 | 0.272 | 0.176 | 0.258 | 0.180 | 0.272 | 0.200 | 0.281 |
| | 192 | 0.222 | 0.300 | 0.220 | 0.291 | 0.223 | 0.302 | 0.283 | 0.360 | 0.245 | 0.319 | 0.240 | 0.300 | 0.249 | 0.325 | 0.252 | 0.320 | 0.247 | 0.311 | 0.241 | 0.302 | 0.248 | 0.315 | 0.252 | 0.312 |
| | 336 | 0.272 | 0.329 | 0.273 | 0.325 | 0.272 | 0.331 | 0.371 | 0.450 | 0.298 | 0.253 | 0.310 | 0.347 | 0.330 | 0.378 | 0.315 | 0.351 | 0.307 | 0.351 | 0.307 | 0.347 | 0.308 | 0.352 | 0.314 | 0.352 |
| | 720 | 0.365 | 0.383 | 0.368 | 0.383 | 0.380 | 0.384 | 0.555 | 0.509 | 0.405 | 0.401 | 0.409 | 0.404 | 0.423 | 0.415 | 0.415 | 0.408 | 0.409 | 0.403 | 0.410 | 0.402 | 0.412 | 0.407 | 0.411 | 0.405 |
| Traffic | 96 | 0.410 | 0.286 | 0.408 | 0.277 | 0.412 | 0.288 | 0.648 | 0.396 | 0.497 | 0.342 | 0.527 | 0.339 | 0.530 | 0.340 | 0.650 | 0.396 | 0.400 | 0.271 | 0.394 | 0.268 | 0.502 | 0.329 | 0.401 | 0.277 |
| | 192 | 0.427 | 0.288 | 0.422 | 0.283 | 0.429 | 0.297 | 0.598 | 0.370 | 0.499 | 0.339 | 0.502 | 0.339 | 0.516 | 0.338 | 0.597 | 0.359 | 0.470 | 0.319 | 0.413 | 0.277 | 0.490 | 0.331 | 0.421 | 0.282 |
| | 336 | 0.439 | 0.305 | 0.436 | 0.295 | 0.445 | 0.306 | 0.605 | 0.373 | 0.520 | 0.349 | 0.510 | 0.327 | 0.533 | 0.343 | 0.605 | 0.362 | 0.489 | 0.333 | 0.428 | 0.283 | 0.512 | 0.341 | 0.434 | 0.389 |
| | 720 | 0.454 | 0.311 | 0.455 | 0.311 | 0.474 | 0.319 | 0.645 | 0.395 | 0.550 | 0.349 | 0.545 | 0.345 | 0.575 | 0.367 | 0.642 | 0.381 | 0.478 | 0.330 | 0.463 | 0.301 | 0.576 | 0.364 | 0.465 | 0.302 |
| Weather | 96 | 0.150 | 0.208 | 0.162 | 0.212 | 0.152 | 0.210 | 0.196 | 0.256 | 0.167 | 0.225 | 0.166 | 0.207 | 0.170 | 0.229 | 0.195 | 0.235 | 0.165 | 0.221 | 0.162 | 0.204 | 0.170 | 0.227 | 0.175 | 0.225 |
| | 192 | 0.194 | 0.251 | 0.207 | 0.251 | 0.196 | 0.254 | 0.238 | 0.299 | 0.208 | 0.263 | 0.216 | 0.253 | 0.211 | 0.270 | 0.240 | 0.270 | 0.212 | 0.261 | 0.213 | 0.252 | 0.214 | 0.270 | 0.225 | 0.257 |
| | 336 | 0.243 | 0.289 | 0.256 | 0.288 | 0.246 | 0.294 | 0.281 | 0.330 | 0.255 | 0.301 | 0.273 | 0.295 | 0.261 | 0.310 | 0.291 | 0.306 | 0.261 | 0.304 | 0.271 | 0.295 | 0.265 | 0.309 | 0.280 | 0.307 |
| | 720 | 0.311 | 0.339 | 0.325 | 0.337 | 0.315 | 0.346 | 0.346 | 0.384 | 0.326 | 0.349 | 0.351 | 0.346 | 0.332 | 0.359 | 0.364 | 0.353 | 0.338 | 0.345 | 0.340 | 0.347 | 0.342 | 0.358 | 0.373 | 0.366 |

Table 10: Detailed results of comparing TIFO and FAN. The best results are highlighted in **bold**. The second best are underlined.

| | | DLinear (Zeng et al., 2023) + TIFO* MSE | MAE | + TIFO MSE | MAE | + FAN MSE | MAE | + RevIN MSE | MAE | iTransformer (Liu et al., 2024b) + TIFO* MSE | MAE | + TIFO MSE | MAE | + FAN MSE | MAE | + RevIN MSE | MAE |
|---|---|---|---|---|---|---|---|---|---|---|---|---|---|---|---|---|---|
| Electricity | 96 | 0.135 | 0.230 | 0.140 | 0.237 | 0.146 | 0.248 | 0.210 | 0.278 | 0.145 | 0.244 | 0.143 | 0.237 | 0.158 | 0.254 | 0.152 | 0.251 |
| | 192 | 0.149 | 0.245 | 0.155 | 0.249 | 0.163 | 0.264 | 0.210 | 0.304 | 0.169 | 0.266 | 0.159 | 0.252 | 0.170 | 0.263 | 0.165 | 0.255 |
| | 336 | 0.165 | 0.262 | 0.171 | 0.267 | 0.180 | 0.282 | 0.223 | 0.309 | 0.178 | 0.271 | 0.172 | 0.266 | 0.183 | 0.281 | 0.180 | 0.272 |
| | 720 | 0.198 | 0.291 | 0.208 | 0.298 | 0.216 | 0.316 | 0.257 | 0.349 | 0.210 | 0.311 | 0.205 | 0.295 | 0.208 | 0.306 | 0.227 | 0.312 |
| ETTh1 | 96 | 0.375 | 0.398 | 0.371 | 0.392 | 0.414 | 0.431 | 0.396 | 0.410 | 0.380 | 0.400 | 0.389 | 0.404 | 0.425 | 0.434 | 0.394 | 0.409 |
| | 192 | 0.410 | 0.417 | 0.404 | 0.412 | 0.446 | 0.451 | 0.445 | 0.440 | 0.429 | 0.427 | 0.447 | 0.440 | 0.483 | 0.468 | 0.460 | 0.449 |
| | 336 | 0.430 | 0.427 | 0.426 | 0.426 | 0.476 | 0.474 | 0.487 | 0.465 | 0.479 | 0.451 | 0.492 | 0.463 | 0.528 | 0.495 | 0.501 | 0.475 |
| | 720 | 0.437 | 0.455 | 0.428 | 0.448 | 0.539 | 0.538 | 0.512 | 0.510 | 0.491 | 0.471 | 0.496 | 0.482 | 0.582 | 0.555 | 0.521 | 0.504 |
| ETTh2 | 96 | 0.273 | 0.335 | 0.273 | 0.336 | 0.316 | 0.376 | 0.344 | 0.397 | 0.298 | 0.352 | 0.297 | 0.345 | 0.358 | 0.408 | 0.300 | 0.349 |
| | 192 | 0.335 | 0.374 | 0.336 | 0.376 | 0.384 | 0.423 | 0.485 | 0.481 | 0.371 | 0.402 | 0.380 | 0.395 | 0.458 | 0.469 | 0.381 | 0.415 |
| | 336 | 0.361 | 0.399 | 0.355 | 0.395 | 0.465 | 0.479 | 0.582 | 0.536 | 0.425 | 0.435 | 0.420 | 0.428 | 0.570 | 0.533 | 0.433 | 0.442 |
| | 720 | 0.388 | 0.429 | 0.384 | 0.423 | 0.671 | 0.596 | 0.836 | 0.659 | 0.420 | 0.444 | 0.410 | 0.432 | 0.786 | 0.651 | 0.426 | 0.445 |
| ETTm1 | 96 | 0.285 | 0.339 | 0.299 | 0.341 | 0.302 | 0.352 | 0.353 | 0.374 | 0.326 | 0.361 | 0.330 | 0.370 | 0.360 | 0.390 | 0.341 | 0.376 |
| | 192 | 0.321 | 0.359 | 0.336 | 0.364 | 0.342 | 0.376 | 0.391 | 0.392 | 0.365 | 0.384 | 0.374 | 0.391 | 0.398 | 0.408 | 0.380 | 0.394 |
| | 336 | 0.355 | 0.380 | 0.370 | 0.383 | 0.385 | 0.402 | 0.423 | 0.413 | 0.395 | 0.403 | 0.408 | 0.414 | 0.435 | 0.436 | 0.419 | 0.418 |
| | 720 | 0.405 | 0.411 | 0.425 | 0.414 | 0.442 | 0.439 | 0.486 | 0.449 | 0.471 | 0.447 | 0.475 | 0.449 | 0.503 | 0.478 | 0.486 | 0.455 |
| ETTm2 | 96 | 0.163 | 0.255 | 0.165 | 0.254 | 0.170 | 0.262 | 0.194 | 0.293 | 0.178 | 0.272 | 0.176 | 0.258 | 0.195 | 0.293 | 0.200 | 0.281 |
| | 192 | 0.222 | 0.300 | 0.220 | 0.291 | 0.230 | 0.306 | 0.283 | 0.360 | 0.247 | 0.311 | 0.241 | 0.302 | 0.276 | 0.350 | 0.252 | 0.312 |
| | 336 | 0.272 | 0.329 | 0.273 | 0.325 | 0.293 | 0.351 | 0.371 | 0.450 | 0.307 | 0.351 | 0.307 | 0.347 | 0.336 | 0.385 | 0.314 | 0.352 |
| | 720 | 0.365 | 0.383 | 0.368 | 0.383 | 0.420 | 0.436 | 0.555 | 0.509 | 0.409 | 0.403 | 0.410 | 0.402 | 0.539 | 0.512 | 0.411 | 0.405 |
| Traffic | 96 | 0.410 | 0.286 | 0.408 | 0.277 | 0.524 | 0.340 | 0.648 | 0.396 | 0.400 | 0.271 | 0.394 | 0.268 | 0.525 | 0.341 | 0.401 | 0.277 |
| | 192 | 0.427 | 0.288 | 0.422 | 0.283 | 0.523 | 0.338 | 0.598 | 0.370 | 0.470 | 0.319 | 0.413 | 0.277 | 0.517 | 0.336 | 0.421 | 0.282 |
| | 336 | 0.439 | 0.305 | 0.436 | 0.295 | 0.537 | 0.343 | 0.605 | 0.373 | 0.489 | 0.333 | 0.428 | 0.283 | 0.530 | 0.339 | 0.434 | 0.389 |
| | 720 | 0.454 | 0.311 | 0.455 | 0.311 | 0.581 | 0.362 | 0.645 | 0.395 | 0.478 | 0.330 | 0.463 | 0.301 | 0.573 | 0.339 | 0.465 | 0.302 |
| Weather | 96 | 0.150 | 0.208 | 0.162 | 0.212 | 0.187 | 0.242 | 0.196 | 0.256 | 0.165 | 0.221 | 0.162 | 0.204 | 0.178 | 0.235 | 0.175 | 0.225 |
| | 192 | 0.194 | 0.251 | 0.207 | 0.251 | 0.227 | 0.280 | 0.238 | 0.299 | 0.212 | 0.261 | 0.213 | 0.252 | 0.220 | 0.275 | 0.225 | 0.257 |
| | 336 | 0.243 | 0.289 | 0.256 | 0.288 | 0.278 | 0.330 | 0.281 | 0.330 | 0.261 | 0.304 | 0.271 | 0.295 | 0.267 | 0.314 | 0.280 | 0.307 |
| | 720 | 0.311 | 0.339 | 0.325 | 0.337 | 0.341 | 0.368 | 0.346 | 0.384 | 0.338 | 0.345 | 0.340 | 0.347 | 0.340 | 0.369 | 0.373 | 0.366 |

## C.2 ANALYSIS RESULTS

### C.2.1 FOURIER DISTRIBUTION ANALYSIS

In the main text we describe using Jensen–Shannon divergence squared ($JSD^2$) and the Kolmogorov–Smirnov (KS) statistic to quantify spectral distribution shift between training and test sets. Here we show another result and the algorithms.

**$JSD^2$ Computation.** For each frequency $\omega_j$, let $a = \{ \text{amp}_{i,j} \mid i \in t_{\text{train}} \}, \quad b = \{ \text{amp}_{i,j} \mid i \in t_{\text{test}} \}$. We build histograms of $a$ and $b$, normalize to probability mass functions, and compute the Jensen–Shannon divergence squared via the Python Library "SciPy" 'jensenshannon' function.

---

**Algorithm 7:** Compute $JSD^2$ between two samples

---

**Input:** Arrays $a, b$; number of bins $B$
**Output:** Jensen–Shannon divergence squared $D_{JSD^2}$
$v_{\min} \leftarrow \min(\min a, \min b)$;
$v_{\max} \leftarrow \max(\max a, \max b)$;
Compute histogram $h_a \leftarrow \text{histogram}(a; B, [v_{\min}, v_{\max}])$;
Compute histogram $h_b \leftarrow \text{histogram}(b; B, [v_{\min}, v_{\max}])$;
Normalize: $p \leftarrow h_a / \sum h_a, \quad q \leftarrow h_b / \sum h_b$;
$D_{JSD} \leftarrow \text{jensenshannon}(p, q, \text{base} = 2)$;
$D_{JSD^2} \leftarrow D_{JSD}^2$;
**return** $D_{JSD^2}$

---

**KS Statistic Computation.** For each frequency index $\omega_j$, using the same samples $a$ and $b$, we compute their empirical cumulative distribution functions (ECDFs) and take the maximum absolute difference.

---

**Algorithm 8:** Compute Kolmogorov–Smirnov statistic

---

**Input:** Arrays $a, b$
**Output:** KS statistic $D_{KS}$
Sort $a \rightarrow a_{\text{sorted}}, \quad$ sort $b \rightarrow b_{\text{sorted}}$;
Let $V \leftarrow \text{unique}(\{a_{\text{sorted}}\} \cup \{b_{\text{sorted}}\})$;
**foreach** $v \in V$ **do**
$\quad F_a(v) \leftarrow \frac{1}{|a|} \left| \{x \in a : x \leq v\} \right|$;
$\quad F_b(v) \leftarrow \frac{1}{|b|} \left| \{x \in b : x \leq v\} \right|$;
$\quad$ Compute $\Delta(v) \leftarrow |F_a(v) - F_b(v)|$;
**end**
$D_{KS} \leftarrow \max_{v \in V} \Delta(v)$;
**return** $D_{KS}$

---

### C.2.2 FOURIER BASIS LEARNING EVALUATION

We include here further 3D views of the ground truth vs. predicted amplitude spectra, sampled from different time-series examples and channels. The forecasting horizon is fixed to 720. For each plot, we pick a specific sample index and a specific channel, to show that our TIFO consistently helps the model recover spectral peaks across the dataset:

- **Compute amplitudes.** For each method tag (`Before`, `FAN`, `TIFO`), we load a 1D slice `data[...,sample,channel]`, take the real FFT via `rfft`, then smooth with `gaussian_filter1d` and clip to $[z_{\min}, z_{\max}]$.
- **Frequency-to-axis mapping.** We linearly map the frequency index range $[f_{\min}, f_{\max}]$ to the extended Y-axis interval, and map amplitude values into the Z-axis range plus a constant offset (`Z_OFFSET`).
- **True vs. predicted curves.** We plot the smoothed true spectrum (in blue) and the predicted spectrum (in red) at a constant $x$-plane (`X_PLANE`), using 3D line plots.

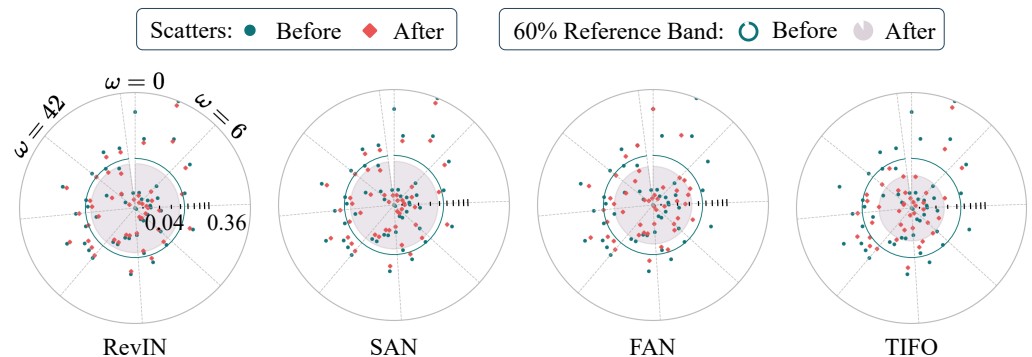

Figure 5: More results on train-test distance compactness: This figure shows another visualization of the JSD$^2$ amplitudes distribution distance between the train and test datasets on the electricity data (on a different channel). Each scatter point represents one frequency component. A smaller radius indicates a smaller distributional gap. Green and red colors represent the results before and after applying the learning method, respectively.

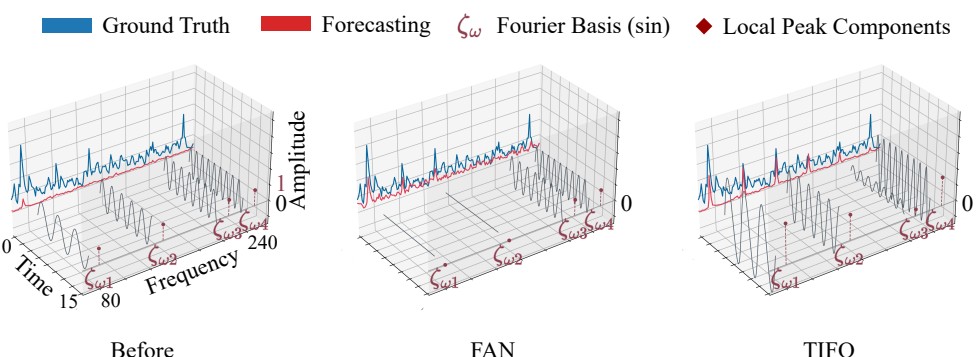

Figure 6: Frequency-domain analysis of Fourier basis Learning. ETTh1 dataset, Channel #1.

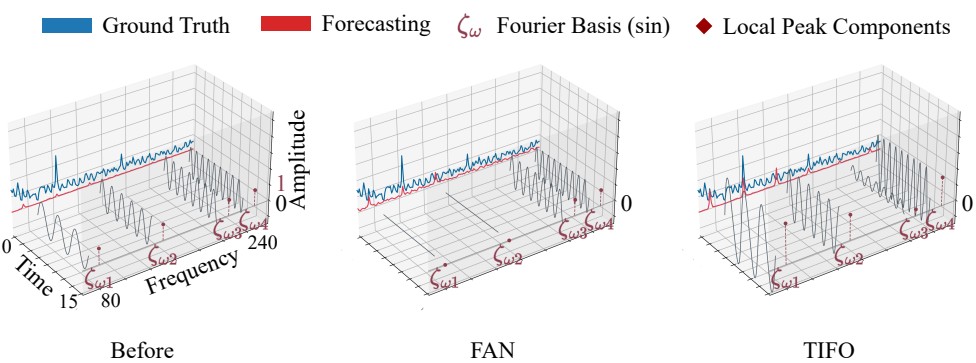

Figure 7: Frequency-domain analysis of Fourier basis Learning. ETTm1 dataset, Channel #2.

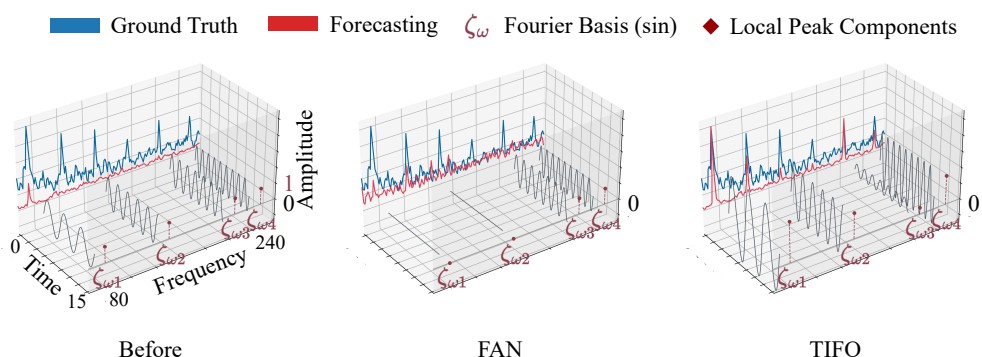

Figure 8: Frequency-domain analysis of Fourier basis Learning. Traffic dataset, Channel #237.

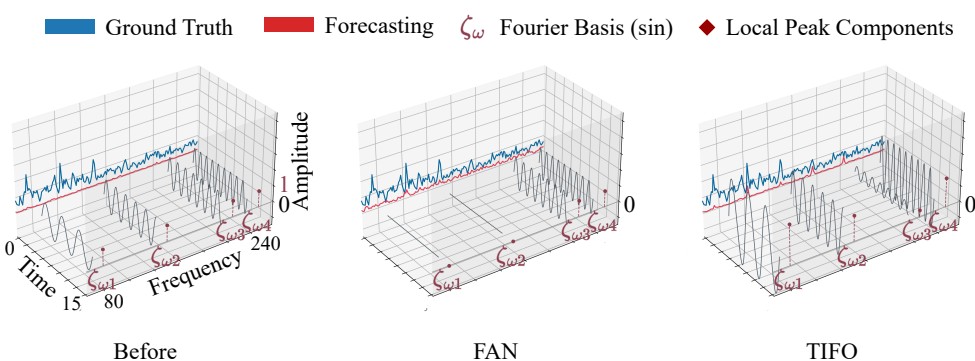

Figure 9: Frequency-domain analysis of Fourier basis Learning. ETTh1 dataset, Channel #6.

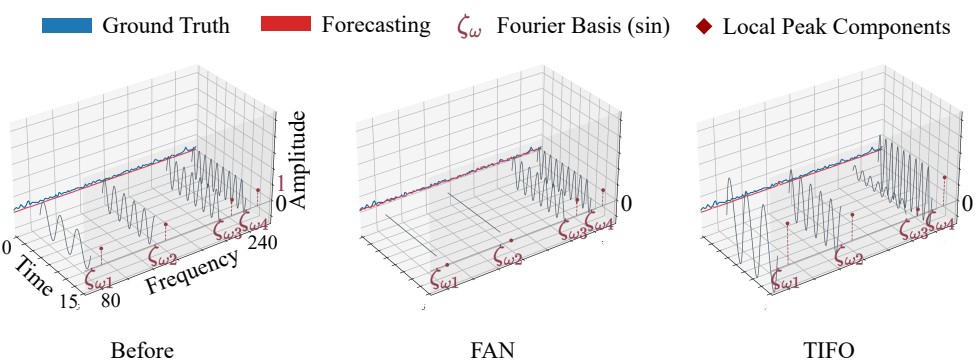

Figure 10: Frequency-domain analysis of Fourier basis Learning. Weather dataset, Channel #7.

## C.3 RUNNING TIME

Table 11 presents the running time comprehensive results of our paper. This table includes the prediction accuracy outcomes on the [Electricity, ETTh1, ETTh2, ETTm1, ETTm2, Traffic, Weather] dataset, utilizing the [DLinear, PatchTST] as the backbone model. We compare our method against SAN across forecasting horizons $H \in \{96, 720\}$.

Table 12: Ablation on window function and FFT resolution on ETTh1 with iTransformer. The **best** results are highlighted.

| Setting | Run 0 | | Run 1 | | Run 2 | | Average | |
|---------|-------|-------|-------|-------|-------|-------|---------|-------|
| | MSE | MAE | MSE | MAE | MSE | MAE | MSE | MAE |
| rect_K48 | 0.4250 | 0.4420 | 0.4020 | 0.4180 | 0.4050 | 0.4210 | 0.4107 | 0.4270 |
| rect_K96 | 0.3902 | 0.4068 | 0.3939 | 0.4105 | 0.3974 | 0.4140 | **0.3938** | **0.4104** |
| hann_K48 | 0.4130 | 0.4260 | 0.4100 | 0.4240 | 0.4120 | 0.4250 | 0.4117 | 0.4250 |
| hann_K96 | 0.4110 | 0.4240 | 0.4090 | 0.4230 | 0.4130 | 0.4260 | 0.4110 | 0.4243 |

Table 13: Effect of scaling the learned TIFO weights at test time by $\alpha$ on ETTh1 with iTransformer. The **best** average performance is highlighted.

| $\alpha$ | H = 96 | | H = 192 | | H = 336 | | H = 720 | | Average | |
|------|-------|-------|-------|-------|-------|-------|-------|-------|-------|-------|
| | MSE | MAE | MSE | MAE | MSE | MAE | MSE | MAE | MSE | MAE |
| 0.00 | 0.4590 | 0.4740 | 0.5170 | 0.5100 | 0.5620 | 0.5330 | 0.5660 | 0.5520 | 0.5260 | 0.5172 |
| 0.25 | 0.4540 | 0.4690 | 0.5120 | 0.5050 | 0.5570 | 0.5280 | 0.5610 | 0.5470 | 0.5210 | 0.5122 |
| 0.50 | 0.4190 | 0.4340 | 0.4770 | 0.4700 | 0.5220 | 0.4930 | 0.5260 | 0.5120 | 0.4860 | 0.4773 |
| 0.75 | 0.3900 | 0.4050 | 0.4440 | 0.4380 | 0.4940 | 0.4650 | 0.4990 | 0.4840 | 0.4567 | 0.4480 |
| 1.00 | 0.3780 | 0.3930 | 0.4460 | 0.4390 | 0.4810 | 0.4520 | 0.4850 | 0.4710 | **0.4475** | **0.4388** |

Table 11: Running time comparison with forecasting lengths $H \in \{96, 720\}$ for all datasets and fixed input sequence length $L = 96$. The **best** results are highlighted.

| Model | Electricity | | ETTh1 | | ETTh2 | | ETTm1 | | ETTm2 | | Traffic | | Weather | |
|-------|-------|-------|-------|-------|-------|-------|-------|-------|-------|-------|-------|-------|-------|-------|
| H | 96 | 720 | 96 | 720 | 96 | 720 | 96 | 720 | 96 | 720 | 96 | 720 | 96 | 720 |
| | DLinear (Zeng et al., 2023) | | | | | | | | | | | | | |
| + TIFO | **16.718** | **27.545** | **3.004** | **3.082** | **2.820** | **3.222** | **13.456** | **13.124** | **11.839** | **12.949** | **23.217** | **43.413** | **13.970** | **16.536** |
| + SAN | 19.159 | 34.914 | 8.688 | 8.054 | 8.373 | 7.811 | 36.706 | 36.564 | 36.620 | 36.550 | 31.378 | 54.518 | 37.739 | 40.731 |
| IMP(%) | **12.8%** | **21.1%** | **65.5%** | **60.5%** | **66.6%** | **58.3%** | **63.1%** | **64.2%** | **67.8%** | **64.3%** | **26.0%** | **20.3%** | **63.0%** | **59.9%** |
| | PatchTST (Nie et al., 2023) | | | | | | | | | | | | | |
| + TIFO | **99.104** | **103.781** | **7.952** | **8.215** | **14.687** | **14.122** | **54.526** | **28.944** | **59.406** | **26.233** | **209.006** | **215.718** | **69.107** | **49.628** |
| + SAN | 313.697 | 322.083 | 16.226 | 15.309 | 16.078 | 14.998 | 63.686 | 65.839 | 65.978 | 66.798 | 550.026 | 557.730 | 81.973 | 81.462 |
| IMP(%) | **68.4%** | **67.7%** | **51.6%** | **46.2%** | **8.00%** | **5.20%** | **14.3%** | **56.0%** | **10.0%** | **60.7%** | **61.6%** | **61.2%** | **15.9%** | **39.8%** |

## C.4 New Experiments Added in the Revision

We have added several new experiments in the revised version to address reviewers' concerns about sensitivity, robustness, and reconstruction strategies.

(1) **Window function and FFT resolution.** On ETTh1 with iTransformer, we study the sensitivity of TIFO to the choice of window function and frequency resolution under the original single global DFT/IDFT setting (no overlap). We compare rectangular vs. Hann windows and $K = 48$ vs. $K = 96$ frequency bins. The results (Table X) show that using the full spectrum with a rectangular window (rect_K96) gives the best performance with small variance across runs, and changing the window or $K$ within a reasonable range does not break the gains of TIFO. We therefore keep the simple global rectangular window with $K = 96$ as the default.

(2) **Robustness–expressiveness trade-off by scaling TIFO weights.** To analyze whether suppressing non-stationary components harms the predictive upper bound, we train TIFO on ETTh1 with iTransformer and, at test time, multiply the learned frequency weights by a scalar $\alpha \in \{1.00, 0.75, 0.50, 0.25, 0.00\}$. Here $\alpha = 0.00$ corresponds to removing TIFO. As shown in Table Y, $\alpha = 1.00$ and $\alpha = 0.75$ achieve the best (and very similar) accuracy; performance starts to degrade at $\alpha = 0.50$, and becomes clearly worse when $\alpha \leq 0.25$ or when TIFO is disabled. This confirms a natural trade-off: fully using the learned TIFO weights improves robustness and accuracy, while aggressively shrinking them harms performance.

Table 14: Comparison of reconstruction strategies (global DFT vs. OLA/WOLA) on ETTh1 with iTransformer. The **best** results are highlighted.

| Setting | Run 0 | | Run 1 | | Run 2 | | Average | |
|---|---|---|---|---|---|---|---|---|
| | MSE | MAE | MSE | MAE | MSE | MAE | MSE | MAE |
| Ori | 0.3902 | 0.4068 | 0.3939 | 0.4105 | 0.3974 | 0.4140 | **0.3938** | **0.4104** |
| hann_50_ola | 0.5887 | 0.5148 | 0.5623 | 0.5053 | 0.5951 | 0.5152 | 0.5820 | 0.5118 |
| sqrt_hann_50_wola | 0.3999 | 0.4135 | 0.3985 | 0.4133 | 0.4032 | 0.4175 | 0.4005 | 0.4148 |
| hann_75_ola | 0.5872 | 0.5130 | 0.5627 | 0.5050 | 0.5944 | 0.5149 | 0.5814 | 0.5110 |

Table 15: Effect of online EMA refresh of frequency masks during inference on ETTh1 with iTransformer. The **best** results are highlighted.

| Setting | Run 0 | | Run 1 | | Run 2 | | Average | |
|---|---|---|---|---|---|---|---|---|
| | MSE | MAE | MSE | MAE | MSE | MAE | MSE | MAE |
| no_update | 0.3902 | 0.4068 | 0.3939 | 0.4105 | 0.3974 | 0.4140 | **0.3938** | **0.4104** |
| ema_0.9 | 0.3916 | 0.4101 | 0.3930 | 0.4105 | 0.3952 | 0.4136 | 0.3932 | 0.4114 |
| ema_0.99 | 0.3903 | 0.4085 | 0.3938 | 0.4110 | 0.3966 | 0.4146 | 0.3936 | 0.4114 |

(3) **Frequency decomposition strategies.** To address phase-distortion and boundary-effect concerns, we compare our original global FFT design with several STFT-style variants on ETTh1 + iTransformer: (i) *Ori* (single global rectangular window, no overlap), (ii) *hann_50_ola* (Hann, 50% overlap, OLA), (iii) *sqrt_hann_50_wola* ($\sqrt{\text{Hann}}$, 50% overlap, WOLA with COLA), and (iv) *hann_75_ola* (Hann, 75% overlap, OLA). Table Z shows that Ori remains the best, the COLA-compliant WOLA variant is slightly worse, and Hann-based OLA variants significantly degrade accuracy. This indicates that, in our setting, the simple global FFT with finite-length periodicity assumption is sufficient, and additional windowing/overlap does not provide benefit.

(4) **Online EMA refresh of stability scores under drift.** We further test whether TIFO requires online updates of the stability-based weights in the presence of distribution drift. During inference on ETTh1 + iTransformer, we re-estimate batch-wise frequency statistics and update the global mask via exponential moving average with three settings: no update (no_update), EMA with decay 0.9, and EMA with decay 0.99. The results (Table W) show that all three settings yield almost identical MSE/MAE, and the small differences are well within run-to-run randomness; EMA(0.9) gives a marginally smaller MSE. This suggests that dataset-level statistics estimated on the training set are already robust, and TIFO does not rely on frequent online re-estimation.

(5) **Sensitivity to STFT hyperparameters (hop size and zero-padding).** Complementary to the global FFT analysis, we also conduct an STFT-style ablation on ETTh1 + iTransformer, varying hop ratio $\in \{0.125, 0.25\}$ and zero-padding ratio $\in \{0, 0.5, 1.0\}$. The results (Table V) show that moderate zero-padding (0.5) yields consistent small gains across hop sizes, while further increasing zero-padding to 1.0 brings negligible improvement. A smaller hop (larger overlap) is slightly better overall, but the effect is modest. Together with the previous experiments, this indicates that TIFO is reasonably robust to FFT-related hyperparameters and that sophisticated STFT configurations are not necessary for the gains reported in the main paper.

# D   LLM USAGE STATEMENT

We used large language models (LLMs) solely for writing support, including grammar correction, sentence refinement, and clarity improvements. All conceptual contributions, algorithm design, code development, experiments, and analyses were conducted entirely by the authors.

Table 16: Sensitivity to STFT hyperparameters (hop ratio and zero-padding) on ETTh1 with iTransformer. The **best** results are highlighted.

| Setting | Run 0 | | Run 1 | | Run 2 | | Average | |
|---|---|---|---|---|---|---|---|---|
| | MSE | MAE | MSE | MAE | MSE | MAE | MSE | MAE |
| hop0.125_zpad0.0 | 0.3950 | 0.4100 | 0.3980 | 0.4130 | 0.3990 | 0.4140 | 0.3973 | 0.4123 |
| hop0.125_zpad0.5 | 0.3920 | 0.4070 | 0.3940 | 0.4090 | 0.3950 | 0.4100 | **0.3937** | **0.4087** |
| hop0.125_zpad1.0 | 0.3930 | 0.4080 | 0.3950 | 0.4100 | 0.3960 | 0.4110 | 0.3947 | 0.4097 |
| hop0.25_zpad0.0 | 0.3970 | 0.4120 | 0.4000 | 0.4150 | 0.4020 | 0.4170 | 0.3997 | 0.4147 |
| hop0.25_zpad0.5 | 0.3950 | 0.4100 | 0.3970 | 0.4120 | 0.3980 | 0.4130 | 0.3967 | 0.4117 |
| hop0.25_zpad1.0 | 0.3940 | 0.4090 | 0.3960 | 0.4110 | 0.3970 | 0.4120 | 0.3957 | 0.4107 |

