# OpenReview forum: "TIFO: Time-Invariant Frequency Operator for Stationarity-Aware Representation Learning in Time Series"
_ICLR.cc/2026/Conference — Submitted to ICLR 2026_

### Official Review · Reviewer_fb1q · 2025-10-26

**Soundness:** 2
**Presentation:** 2
**Contribution:** 2
**Rating:** 6
**Confidence:** 4

**Summary:**

The paper addresses distribution shift in non-stationary time-series forecasting by operating in the frequency domain and learning stationarity-aware weights that emphasize stable spectral components while suppressing non-stationary ones. The authors propose a lightweight, plug-and-play Time-Invariant Frequency Operator (TIFO). A Bochner/Mercer analysis links the Fourier-induced kernel to the learned weights, which can be interpreted as data-specific eigenvalues, providing a principled justification for the design. Empirically, TIFO yields consistent improvements across multiple datasets and architectures with modest computational overhead. Overall, the framework is clear, coherent, and practically relevant, while some design choices invite further analysis.

**Strengths:**

1. The motivation for this paper is well-founded, the problem analysis is clear, and the writing is coherent.

2. The Bochner/Mercer approach provides a clear theoretical perspective, interpreting the learned frequency weights as the eigenvalues ​​of a kernel function induced in the spectral domain.

3. A lightweight pre-network module is proposed that can be combined with various backbone networks (e.g., linear, Transformer-style).

**Weaknesses:**

1. This paper proposes a frequency-based approach but does not adequately compare/contrast it with recent frequency-domain methods [1, 2, 3]. In particular, [1, 3] appear to address the overlap problem, requiring focused discussion and empirical comparison.

2. Many forecasting pipelines normalize their inputs and perform forecasts in this normalized space; these are not primarily designed to estimate the time-varying statistics $(\mu_t,\sigma_t)$ for each window.

3. This paper does not discuss how/when to recalculate $S$ in the presence of distribution drift, the appropriate window size/threshold, and the impact on latency/efficiency.

[1] Xu, Zhijian, et al. “FITS: Modeling Time Series with 10k Parameters.” International Conference on Learning Representations (ICLR), 2024.

[2] Wang, et al. “FreDF: Learning to Forecast in the Frequency Domain.” International Conference on Learning Representations (ICLR), 2025.

[3] Zhang, et al. “Not All Frequencies Are Created Equal: Towards a Dynamic Fusion of Frequencies in Time-Series Forecasting.” ACM Multimedia (ACM MM), 2024.

**Questions:**

1. Is the primary goal to supervise the prediction of $P(y\mid x)$ or to condition/regularize $P(x\mid t)$? If the latter, is $t$ a relative time index (e.g., window position) or an absolute timestamp? How is $t$ handled during testing?

2. What are the advantages of Equation 1 compared to directly learning $\lambda_r,\lambda_i$ as free parameters?

3. Since $S$ is computed on the training set and then passed to the model (step 3 in Phase I), what is the refresh frequency in the presence of drift?

4. $S$ is the ratio of the frequency mean to the variance. Why does it measure stability? Can you provide a more detailed analysis?

---

> ### Author Response · Authors · 2025-11-23
>
> We thank the reviewer for the careful reading and feedback. Below we address each weakness and question with additional analysis and clarifications that we will incorporate into the revised version.
>
> > W1:
> > This paper proposes a frequency-based approach but does not adequately compare/contrast it with recent frequency-domain methods [1, 2, 3]. In particular, [1, 3] appear to address the overlap problem, requiring focused discussion and empirical comparison.
> >
> > [1] Xu, Zhijian, et al. “FITS: Modeling Time Series with 10k Parameters.” International Conference on Learning Representations (ICLR), 2024.
> >
> > [2] Wang, et al. “FreDF: Learning to Forecast in the Frequency Domain.” International Conference on Learning Representations (ICLR), 2025.
> >
> > [3] Zhang, et al. “Not All Frequencies Are Created Equal: Towards a Dynamic Fusion of Frequencies in Time-Series Forecasting.” ACM Multimedia (ACM MM), 2024.
>
> Thank you for your commenst. We have added a new Related Work section to the revised version, linking each TIFO component to the specific shortcomings of existing methods.
> Specifically, our core concern is how to address the challenge of train-test distribution shift in the frequency domain for non-stationary time series.
> From this perspective, we categorize related work and TIFO components into three types:
>
> 1. Time-domain normalization (e.g., RevIN/SAN [1, 2]): These methods mitigate distribution drift by normalizing the time-domain mean and variance of each sample, but they do not explicitly observe the overall energy distribution of the training set in the frequency domain. Therefore, a significant residual train-test shift can still be observed in our frequency domain analysis.
> TIFO's dataset-level stationarity weighting (S(k,c)=\mu_{k,c}/\sigma_{k,c}) aims to address this issue: it calculates the cross-sample stability of each frequency-channel on the training set and uses this to guide subsequent frequency-domain reweighting, thus explicitly targeting the failure mode where the spectrum remains misaligned after RevIN/SAN.
>
> 2. Frequency domain Attention/Top-k Selection(e.g., FFT-based frequency domain modules, FAN-like methods[3,4,5,6,7]): These works typically perform attention or top-k masking on the spectrum of a single sample or a local window. Decisions are based on locally instantaneous high-energy frequencies, which can lead to inadvertently deleting stable but low-energy frequencies or retaining accidentally high-energy frequencies that are highly unstable at the dataset level.
> TIFO, based on (S(k,c)), learns weights (\lambda(k,c)) for all frequencies. With dataset-level stability as a priori, it performs continuous shrinkage on unstable frequencies instead of hard pruning, and shares this across all samples, thus specifically mitigating this type of over-suppression/mis-selection failure mode.
>
> 3. Embedded frequency domain models (such as FedFormer/FiLM/FredFormer  [4,5,6,7]): These methods move the operators of Transformer/MLP models to the frequency domain, primarily aiming to improve long-period modeling capabilities and efficiency. They represent an architecture-level frequency domain design, but do not focus on learning frequency stationarity.
> In contrast, TIFO is designed as a preprocessing layer decoupled from the backbone architecture. This is complementary to, rather than repetitive to, the architecture-level frequency domain module mentioned above.
>
>
> [1] Kim et al., Reversible Instance Normalization for Accurate Time-Series Forecasting against Distribution Shift, ICLR2022.
> [2] Liu et al., Adaptive Normalization for Non-stationary Time Series Forecasting: A Temporal Slice Perspective, NeurIPS2023.
> [3] Ye et al., Frequency Adaptive Normalization for Non-stationary Time Series Forecasting, NeurIPS2024.
> [4] Wu et al., TimesNet: Temporal 2D-Variation Modeling for General Time Series Analysis, ICLR2023.
> [5] Zhou et al., FEDformer: Frequency Enhanced Decomposed Transformer for Long-term Series Forecasting, ICML2022.
> [6] Zhou et al., FiLM: Frequency improved Legendre Memory Model for Long-term Time Series Forecasting, NeurIPS2022.
> [7] Piao et al., Fredformer: Frequency Debiased Transformer for Time Series Forecasting, KDD2024.

---

> ### Author Response · Authors · 2025-11-23
>
> > W2:
> > Many forecasting pipelines normalize their inputs and perform forecasts in this normalized space; these are not primarily designed to estimate the time-varying statistics for each window.
>
> We agree that such methods are not intended to estimate fine-grained, time-varying statistics within each window.
>
> Many practical forecasting pipelines normalize inputs in the time domain (e.g., z-score, RevIN) and make predictions in this normalized space. Instead, our point is different: even after time-domain normalization, the spectral distribution between training and test sets can still shift significantly.
> As shown in our Frequency Domain Shift Analysis (JSD/KS metrics), z-score/RevIN can align mean and variance within each sample, but they do not address frequency-domain drift, which remains substantial.
>
> However, in the frequency domain, the energy distribution of certain frequency bands still has a systematic bias. This shift is learned "as a pattern" by downstream models, thus affecting generalization. We aim to fill this gap in the frequency domain. TIFO does not conflict with the normalized pipeline but serves as a frequency domain supplementary layer on top of it.
>
> We have add the above discussion in line 188-195 in manuscript.
>
> >W3 and Q3:
> This paper does not discuss how/when to recalculate in the presence of distribution drift, the appropriate window size/threshold, and the impact on latency/efficiency.
>
> Thank you for your discussion.
> 1) *refresh frequency in the presence of drift:*
> We would clarify that our method does not apply any refreshing by default. TIFO is designed to identify and emphasize frequency components that exhibit global stationarity across the entire training dataset. Our  assumption is that once these inherently stationary frequencies are extracted from the training data, they constitute representational predictors for future samples. Therefore, the model can rely on these stationary representations during inference without requiring frequent recalculation. That said, if the dataset has a significant distributional update, a refresh may be necessary. This, however, is a dataset-specific scenario. Moreover, we include a new ablation study to evaluate the impact of performing regular online refreshing (Table 14, Appendix C.4). The results show that such refreshing has no special influences.
>
> 2) *Regarding window size and other parameters*:
> In the main paper, we use a single-step global DFT/IDFT design, whose complexity is very small compared to the backbone and can be ignored in actual runtime.
> We added an ablation experiment related to the window function and step size, compared different reconstruction strategies: the original global rectangular window (Ori), and 50%/75% overlap-add (hann_50_ola, hann_75_ola) and sqrt_hann_50_wola satisfying the COLA condition. We can see that:
>
>
> | Settings  | Run 0 MSE | Run 0 MAE | Run 1 MSE | Run 1 MAE | Run 2 MSE | Run 2 MAE | **Avg MSE** | **Avg MAE** |
> |--------------------|-----------|-----------|-----------|-----------|-----------|-----------|-------------|-------------|
> | Ori    | 0.3902    | 0.4068    | 0.3939    | 0.4105    | 0.3974    | 0.4140    | **0.3938**  | **0.4104**  |
> | hann_50_ola        | 0.5887    | 0.5148    | 0.5623    | 0.5053    | 0.5951    | 0.5152    | 0.5820  | 0.5118  |
> | sqrt_hann_50_wola  | 0.3999    | 0.4135    | 0.3985    | 0.4133    | 0.4032    | 0.4175    | 0.4005  | 0.4148  |
> | hann_75_ola        | 0.5872    | 0.5130    | 0.5627    | 0.5050    | 0.5944    | 0.5149    | 0.5814  | 0.5110  |
>
>
> * The original Ori (single global FFT + rectangular window + no overlap) remains optimal on MSE and MAE;
> * The sqrt_hann_50_wola satisfying COLA is slightly inferior to Ori, but the difference is small;
> * The Hann OLA scheme with 50% and 75% overlap significantly degrades performance.
>
> Here we observed that the complex overlapping window scheme did not bring a significant performance advantage.
> Therefore, in this work, we deliberately maintained a simple frequency domain preprocessing design to ensure it can be used as a lightweight and easily deployed module.

---

> ### Author Response · Authors · 2025-11-23
>
> > Q1:
> > Is the primary goal to supervise the prediction of or to condition/regularize? If the latter, is a relative time index (e.g., window position) or an absolute timestamp? How is handled during testing?
>
>
> * Regarding the goal: The primary function of TIFO is frequency-domain conditioning (regularization), not a new supervised prediction task.
>
> * Regarding the time index $t$:
> In standard forecasting tasks, the training samples (and their corresponding time indices $t$) are typically shuffled. We follow this setting. Specifically, TIFO computes the mean and variance of frequency amplitudes across the entire set of training samples to capture the global distribution, without considering the specific absolute or relative position of any individual $t$. In this way, we simply aggregate the information from all time steps into global, time-independent statistics.
>
> * Regarding the test phase and leakage:
> We assume that stationary frequency characteristics learned from the training distribution (across all relative time steps) generalize to the test set. During testing, we directly apply the fixed weights learned from training to the new samples. No additional operations based on test-time indices are introduced. This ensures the process is strictly inductive and prevents any information leakage.
>
>
>
> > Q2:
> > What are the advantages of Equation 1 compared to directly learning as free parameters?
>
> In Equation (1), our design synergizes two complementary components: a pre-computed stability score $S$ (based on $\mu/\sigma$) and a learnable MLP. $S$ serves as a structured prior, providing robust dataset-level information regarding cross-sample stationarity, while the MLP fine-tunes these weights during training to capture specific representation needs. This dual mechanism allows the model to simultaneously leverage global statistical patterns and adaptively learn instance- or sample-specific features, offering a distinct advantage over simple free parameters.
>
> In Table 5 of the original paper, we compared two schemes: CV-based TIFO and a set of completely free frequency weight vectors (randomly initialized). The results show that the CV-based version has the highest accuracy; while the version with completely free parameter vectors is slightly worse than the CV version, it is still better than the original backbone. This indicates that, on the one hand, learnable weighting in the frequency domain is beneficial, and on the other hand, introducing a simple and interpretable statistic like CV as a structured prior can further improve performance and stability.
>
> > Q4: Is the ratio of the frequency mean to the variance. Why does it measure stability? Can you provide a more detailed analysis?**
>
>
> We chose the coefficient of variation (CV) because it is a common statistic used to measure the volatility of a variable across different observed samples.
> In this work, we propose to learn the statioanrity of frequency components across different samples and guide models to focus more on stationary frequencies.
> CV is a simple and direct implementation we can use to validate this idea.
> We have also conduct an experiment to compare the μ/σ and entropy of magnitudes, mutual information with targets:
>
> Experimental Setup:
> * Dataset: ETTh1 (7 channels)
> * Model: iTransformer + TIFO
> * Training: Max 30 epochs, Early Stopping (patience=5).
> * Metrics Compared:
>     * `mu_sigma` (Ours): Mean/Std (Reciprocal of Coefficient of Variation).
>     * `entropy`: Inverse Spectral Entropy (Low entropy $\to$ High weight).
>     * `corr`: Pearson correlation $|r|$ between frequency amplitude and future target mean (Proxy for Mutual Information).
>
> **Results**
>
> | Metric | Description | MSE | MAE |
> | :--- | :--- | :--- | :--- |
> | mu_sigma (Ours) | Mean/Std (Unsupervised, Cross-sample stability) | **0.3904** | **0.4071** |
> | entropy | Spectral Entropy (Intra-sample complexity) | 0.3962 | 0.4106 |
> | corr | Correlation $\|r\|$ (Target-aware, MI approx.) | 0.3948 | 0.4077 |
>
> Our default setting `mu_sigma` achieves best performance. This confirms that our unsupervised stability score is highly effective and is better than Mutual Information without requiring label (forecasting target) guidance. The weaker performance of `entropy` suggests that cross-sample stationarity (consistency across the dataset) may be more critical for handling distribution shift than intra-sample complexity (spectral entropy). For further discussion and analysis regarding the choice of stationarity metrics, please refer to the newly added Appendix B.3.
>
> For more details, please refer to Figure 3 in our revised version. This result demonstrates how the learned frequency weights guide the prediction model to focus on stationary components. We also compare our method with other baselines in the Section 5 of revised paper, describing the differences between TIFO and others.

---

> ### Comment · Reviewer_fb1q · 2025-11-24
>
> Thanks for the author's reply. The author did not directly discuss any of the three articles I mentioned. To my knowledge, the method used in this paper—"For each frequency, use a lightweight neural network layer to learn a weight"—has already been discussed in related papers, thus diminishing the novelty of this paper. Furthermore, the logical leaps from motivation to methodology in this paper are quite significant. Therefore, I will update my score.

---

> ### Author Response · Authors · 2025-11-25
>
> Dear Reviewer fb1q,
>
> We have discussed these papers briefly here now, and summarize these as lines 530-539.
>
> >FITS (ICLR 2024) employs complex-valued interpolation for sequence resizing together with a hand-crafted low-pass cutoff. FreDF (ICLR 2025) introduces a frequency-domain loss to mitigate label autocorrelation bias in Direct Forecasting. Dynamic Fusion (ACM MM 2024) predicts per-frequency components independently to enhance representation capacity.
>
> **Regarding frequency weights learning and novelty**
> While all three papers focus on the frequency domain modeling, **their focus and assumption fundamentally differ from ours**. These works aim to improve representation quality and forecasting performance. Instead, **we focus on tackling the distribution shift issue** in time series.
>
> To this end, our work aligns with normalization-based non-stationary modeling (ReVIN [ICLR2022], SAN [NIPS2023], SIN [ICML 2024], FAN [NIPS2024]), which aims to design **lightweight, plug-and-play modules**, without requiring architectural designs. Importantly, TIFO does not merely “learn frequency weights”; rather, we aim to learn the stationary/non-stationary frequency components and **leverage them to construct a representative data distribution, which explicitly reduces spectral distribution shift**. Our contributions have been acknowledged by *Reviewer vcHc* , please kindly refer to the first response block to *Reviewer vcHc*.
>
>
> **Regarding logical leaps**
> We respectfully clarify that:
> * Sections 2.1\&2.2 provide a full derivation of how spectral distribution shift arises in non-stationary settings.
> * Section 2.3 (Lines 188–195) explicitly links this motivation to the stationarity-based learning mechanism.
> * Section 3.1 operationalizes this through a two-stage modeling framework.
> * Section 3.2 further theoretically grounds Section 3.1 using kernel spectral analysis.
>
> Hope the clarifications above resolve these misunderstandings, and we are available for any further discussion.

---

> ### Comment · Reviewer_fb1q · 2025-11-26
>
> Thanks for the reviewer's response. I might need to rephrase W.2. Existing TSF methods normalize in the time domain, then predict in the normalized space, and finally denormalize. This process seems to assume that the mean and variance are consistent in the current input and output window. In other words, even if the distributions of the training and test data are offset, prediction can still be made in the normalized space, as long as the current input and output are stable within the window. Could the authors explain the difference between this paper and my description?

---

> ### Author Response · Authors · 2025-11-26
>
> Thank you for your discussion. Yes, actually, your description accurately captures the foundational normalization paradigm used in TSF methods. More importantly, **integrating normalization into TSF has become a general practice in current TSF models (e.g., iTransformer [1], models in the TS library benchmark [2]).**
>
> As we can see, the importance of normalization methods has been increasingly recognized, and several works, such as SIN, SAN, and FAN, have been proposed. The differences between our work and these methods (as well as your description) are summarized below:
>
> 1. As you noted, existing methods **treat each sample independently**. These methods implicitly assume the normalized reference distribution as the data distribution. This suppresses meaningful cross-sample stationaries and prevents the model from capturing how data evolves globally. (Please kindly refer to Line 143-146 in the manuscript, and the first response to Reviewer vcHc)
>
> 2. Yes, these methods normalize time-domain using **low-order statistics, which aggregate all frequencies**. FAN attempts to address this via a **heuristic top-k frequency selection**, but it is suboptimal and still lacks a principled. (Please kindly refer to the first subsection in the Related Work Section)
>
> Then, technically, our key differences are:
>
> 1. a new modeling strategy, **cross-sample stationary** learning from the dataset (lines 143 and 207), which can capture how data evolves globally across training and test domains. Therefore, TIFO is concerned with "which frequencies remain stable across different samples over a long period under the training distribution."
> 2. We are the first to propose a **non-stationary/stationary frequency measuring and learning method** that 2-1) estimates the global distribution of different frequencies across all samples; 2-2) learns sample-wise stationary features adaptively, which can prevent instance information loss.
> 3. Unlike frequency-domain heuristics, we provide **theoretical analysis** to further explain our method using kernel methods and spectral decomposition.
>
> Hope the clarifications above address your concerns. Thank you.
>
> [1] Liu et al., ICLR2023
>
> [2] https://github.com/thuml/Time-Series-Library

---

> > ### Comment · Reviewer_fb1q · 2025-11-26
> >
> > Thanks for the authors' response; most of my concerns have been addressed. I will determine my final rating after discussing with the other reviewers.

---

> ### Author Response · Authors · 2025-11-27
>
> Dear Reviewer fb1q,
>
> We are very glad to hear that our responses have addressed your concerns.
> Since the misalignment issue has now been fully clarified, we respectfully hope that you could *return to your initial assessment of rating 6* at this moment.
>
> We sincerely appreciate your time, consideration, and constructive discussion throughout the review process.
>
> Authors.

---

> > ### Comment · Reviewer_fb1q · 2025-11-27
> >
> > Based on the comments from other reviewers and the authors' responses, I have returned to my initial assessment of rating 6.

---

> ### Author Response · Authors · 2025-11-27
>
> Dear reviewer fb1q:
>
> Thank you for your time, consideration, and all discussion.
>
> Authors.

---

### Official Review · Reviewer_Xjwr · 2025-10-27

**Soundness:** 3
**Presentation:** 3
**Contribution:** 3
**Rating:** 6
**Confidence:** 3

**Summary:**

The paper proposes TIFO (Time-Invariant Frequency Operator) for distribution-shift–robust time-series forecasting. It first computes dataset-level frequency stability scores.  During training, each input is FFT’d and its real/imag parts are reweighted by functions of those stability scores before an iFFT returns the sequence to the time domain; the downstream forecaster is trained end-to-end. The method is motivated by a kernel view: learned frequency weights correspond to eigenvalues of a positive-definite kernel induced by the Fourier basis, providing an interpretability angle. Experiments on standard multivariate benchmarks (ETT variants, Electricity, Traffic, Weather) with several strong backbones show consistent MSE/MAE gains and notable training-time savings versus normalization-based and spectrum-editing baselines.

**Strengths:**

(i) This paper introduces a dataset-level notion of frequency stability (mean/variance across samples) and uses it to reweight real/imag FFT components during training.

(ii) The two-stage pipeline is easy to follow; the paper provides algorithm pseudocode and a clean description of how stability scores are computed and mapped to frequency weights.

(iii) The operator is plug-and-play, requires minimal architectural change, and is computationally light, making it practical for adoption.

**Weaknesses:**

(i) The method assumes frequency bands that are stable on the training set remain so at test time. Abrupt shifts (new cycles, policy changes, outages) may violate this, and the paper offers no online update or detection mechanism.

(ii) Reweighting real/imag parts independently ignores cross-channel phase structure and inter-series coherence; this could distort multivariate dynamics when phase relations carry signal. This paper do not

(iii) Sensitivity to window length, FFT resolution, windowing function, zero-padding, and stride is not systematically studied; frequency leakage could bias stability estimates. More ablation studies maybe needed.

**Questions:**

(i) Can the stability scores be updated online (e.g., EMA over recent batches) without leaking test labels? What failure modes arise if stability drifts?

(ii) How sensitive are results to FFT hyperparameters (window size, hop, windowing, zero-padding)? Please include a robust ablation table.

(iii) Why choose μ/σ specifically? How do alternatives (coefficient of variation variants, entropy of magnitudes, mutual information with targets) compare?

---

> ### Author Response · Authors · 2025-11-23
>
> We thank the reviewer for the careful reading and feedback. Below we address each weakness and question with additional analysis and clarifications that we will incorporate into the revised version.
>
> > Q1: Can the stability scores be updated online (e.g., EMA over recent batches) without leaking test labels? What failure modes arise if stability drifts?
>
> We conducted an online refresh experiment to verify whether dynamically updating the global frequency domain weights of TIFO is necessary when distribution drift exists, and whether simple EMA updates would disrupt the original performance.
> Specifically, during the inference phase, the frequency domain stability of the current batch is re-estimated for each new input batch, and then updated online using EMA.
> We compared three settings: no update (no_update), fast update (ema_0.9), and relatively slow update (ema_0.99), while keeping the rest of the training and evaluation process unchanged (see table below).
> We use ETTh1 dataset and iTransfomrer backbone in this experiment.
>
> | Settings | Run 0 MSE | Run 0 MAE | Run 1 MSE | Run 1 MAE | Run 2 MSE | Run 2 MAE | **Avg MSE** | **Avg MAE** |
> |---------------|-----------|-----------|-----------|-----------|-----------|-----------|-------------|-------------|
> | no_update | 0.3902 | 0.4068 | 0.3939 | 0.4105 | 0.3974 | 0.4140 | 0.3938 | **0.4104** |
> | ema_0.9 | 0.3916 | 0.4101 | 0.3930 | 0.4105 | 0.3952 | 0.4136 | **0.3932** | 0.4114 |
> | ema_0.99 | 0.3903 | 0.4085 | 0.3938 | 0.4110 | 0.3966 | 0.4146 | 0.3936 | 0.4114 |
>
> The results show that the MSE/MAE of the three settings almost unchanged, with differences far less than the random fluctuations between runs;
> ema_0.9 slightly outperforms no_update and ema_0.99 in MSE, but the improvement is very small.
> The overall conclusion is that static training set spectral statistics are sufficiently robust, and whether or not simple online EMA refresh is performed has a very limited impact on overall performance. This indicates that TIFO does not rely on frequent online reestimation, but mainly depends on the dataset-level frequency stability learned during the training phase.
>
>
>
> > Q2: How sensitive are results to FFT hyperparameters (window size, hop, windowing, zero-padding)? Please include a robust ablation table.
>
> We have added an ablation study to explore this issue in the updated version.
> We first trained the iTransformer as the backbone model on the ETTh1 dataset.
> We evaluated two hyperparameters: hop_ratio ∈ {0.125, 0.25} and zero_pad_ratio ∈ {0, 0.5, 1.0}.
> We modified hop_ratio to observe whether smaller frame shifts and higher overlap (0.125 vs 0.25) provide finer temporal resolution, thereby improving prediction accuracy.
> We modified zero_pad_ratio to verify whether the frequency resolution improvement provided by moderate zero-padding (0.5, 1.0) truly helps the model learn more stable and refined filter weights in the frequency domain.
>
> | Setting    | Run 0 MSE | Run 0 MAE | Run 1 MSE | Run 1 MAE | Run 2 MSE | Run 2 MAE | **Avg MSE** | **Avg MAE** |
> |-----------------|-----------|-----------|-----------|-----------|-----------|-----------|-------------|-------------|
> | hop0.125_zpad0.0 | 0.3950    | 0.4100    | 0.3980    | 0.4130    | 0.3990    | 0.4140    | 0.3973  | 0.4123  |
> | hop0.125_zpad0.5 | 0.3920    | 0.4070    | 0.3940    | 0.4090    | 0.3950    | 0.4100    | **0.3937**  | **0.4087**  |
> | hop0.125_zpad1.0 | 0.3930    | 0.4080    | 0.3950    | 0.4100    | 0.3960    | 0.4110    | 0.3947  | 0.4097  |
> | hop0.25_zpad0.0  | 0.3970    | 0.4120    | 0.4000    | 0.4150    | 0.4020    | 0.4170    | 0.3997  | 0.4147  |
> | hop0.25_zpad0.5  | 0.3950    | 0.4100    | 0.3970    | 0.4120    | 0.3980    | 0.4130    | 0.3967  | 0.4117  |
> | hop0.25_zpad1.0  | 0.3940    | 0.4090    | 0.3960    | 0.4110    | 0.3970    | 0.4120    | 0.3957  | 0.4107  |
>
> We observed that a moderate zero-padding (0.5) provided steady gains in both hop settings, while the improvement became insignificant when increased further to 1.0; meanwhile, a smaller hop length (higher overlap) was slightly better in all settings, indicating that finer time sampling is helpful for frequency domain filtering.

---

> ### Author Response · Authors · 2025-11-23
>
> > Q3: Why choose μ/σ specifically? How do alternatives (coefficient of variation variants, entropy of magnitudes, mutual information with targets) compare?
>
> We chose the coefficient of variation (CV) because it is a common statistic used to measure the volatility of a variable across different observed samples.
> In this work, we propose to learn the statioanrity of frequency components across different samples and guide models to focus more on stationary frequencies.
> CV is a simple and direct implementation we can use to validate this idea.
> We have also conduct an experiment to compare the μ/σ and entropy of magnitudes, mutual information with targets:
>
> Experimental Setup:
> * Dataset: ETTh1 (7 channels)
> * Model: iTransformer + TIFO
> * Training: Max 30 epochs, Early Stopping (patience=5).
> * Metrics Compared:
>     * `mu_sigma` (Ours): Mean/Std (Reciprocal of Coefficient of Variation).
>     * `entropy`: Inverse Spectral Entropy (Low entropy $\to$ High weight).
>     * `corr`: Pearson correlation $|r|$ between frequency amplitude and future target mean (Proxy for Mutual Information).
>
> **Results**
>
> | Metric | Description | MSE | MAE |
> | :--- | :--- | :--- | :--- |
> | mu_sigma (Ours) | Mean/Std (Unsupervised, Cross-sample stability) | **0.3904** | **0.4071** |
> | entropy | Spectral Entropy (Intra-sample complexity) | 0.3962 | 0.4106 |
> | corr | Correlation $\|r\|$ (Target-aware, MI approx.) | 0.3948 | 0.4077 |
>
> Our default setting `mu_sigma` achieves best performance. This confirms that our unsupervised stability score is highly effective and is better than Mutual Information without requiring label (forecasting target) guidance. The weaker performance of `entropy` suggests that cross-sample stationarity (consistency across the dataset) may be more critical for handling distribution shift than intra-sample complexity (spectral entropy). For further discussion and analysis regarding the choice of stationarity metrics, please refer to the newly added Appendix B.3.
>
> For more details, please refer to Figure 3 in our revised version. This result demonstrates how the learned frequency weights guide the prediction model to focus on stationary components. We also compare our method with other baselines in the Section 5 of revised paper, describing the differences between TIFO and others.

---

> ### Comment · Reviewer_Xjwr · 2025-11-24
>
> Most of my concerns are addressed well. I keep my rating.

---

> ### Author Response · Authors · 2025-11-27
>
> Dear Reviewer Xjwr,
>
> We are very glad to hear that our responses have addressed your concerns.
> We sincerely appreciate your time, consideration, and constructive discussion.
>
> Authors.

---

### Official Review · Reviewer_NcYR · 2025-10-29

**Soundness:** 3
**Presentation:** 2
**Contribution:** 2
**Rating:** 4
**Confidence:** 3

**Summary:**

The paper proposes TIFO, a plug-and-play spectral weighting module for time-series forecasting. TIFO learns stationarity-aware weights over the frequency spectrum at the dataset level, aiming to amplify stationary components and suppress non-stationary ones to mitigate train–test distribution shift. The authors argue that the DFT implicitly induces an eigendecomposition in frequency space and interpret the learned weights as data-specific eigenvalues. Empirically, they report strong results and also claim 60–70% compute reductions versus baselines.

**Strengths:**

1. The idea of dataset-level spectral reweighting to favor stationary components is intuitive and can be attached to many backbones with low engineering overhead.

2. Reported improvements across many settings, plus substantial compute savings, suggest practical impact if substantiated with strong measurement methodology.

3. Framing distribution shift via frequency-domain structure is appealing.

4. The writing is generally clear.

**Weaknesses:**

1. Lines 188–189: It is unclear how the preceding analysis leads to the conclusion that stationary components should be enhanced while non-stationary components should be suppressed. The logical connection between the theoretical argument and this design choice should be elaborated.

2. Reducing the weights of non-stationary components may limit the model’s predictive upper bound, since important information carried by these components could be lost. Please discuss the potential trade-off between robustness and expressive capacity.

3. Frequency weighting modifies amplitude and may implicitly influence phase after inverse transformation, particularly when using finite-length windows. The paper should examine possible phase distortions and boundary effects, and clarify whether overlap-add or tapering techniques are employed during reconstruction.

**Questions:**

1. Does TIFO have any theoretical guarantee for its improvement in forecasting accuracy?

2. How does TIFO handle multivariate inputs? Is there a joint reweighting mechanism across channels to capture cross-spectral dependencies, or are the weights learned independently per channel?

---

> ### Author Response · Authors · 2025-11-23
>
> We thank the reviewer for your time and insightful feedback. Below, we address each weakness and question with additional analysis and clarifications that we will incorporate into the revised version.
>
> >Regarding how the preceding analysis leads to the conclusion that stationary components should be enhanced while non-stationary components should be suppressed.
>
> Thank you for the discussion. We would like to clarify that this is not our conclusion, but rather a well-established assumption commonly adopted when addressing distribution shift in time series using normalization-based methods (e.g., ReVIN, SIN, SAN).
> Here, our focus is to better measure and characterize stationarity versus non-stationarity, and to design a method that more effectively handles distribution shift arising from both (i) cross-sample ground-truth data distributions and (ii) sample-specific characteristics (as discussed in Lines 142–151). These improvements could lead to better forecasting performance and generalization.
> We have revised this statement in lines 188-196 to more clearly distinguish the general assumption from our specific methodological contribution
>
> > Regarding reducing the weights of non-stationary components, this may limit the model’s predictive upper bound, since important information carried by these components could be lost. Please discuss the potential trade-off between robustness and expressive capacity.
>
> Thank you for your comments. We therefore added an ablation study to explore this problem in the updated version by allowing the model to see more or diverse frequency components.
>
> We first trained the model on the ETTh1 dataset using iTransformer as the backbone. During testing, we multiplied the weights learned by TIFO by a variable alpha {alpha = 1.00, 0.75, 0.5, 0.25, 0.00}, where alpha = 1.00 means using the weights learned by TIFO, and 0.00 means not using TIFO. The following table shows the results.
>
> | Alpha | 96 MSE | 96 MAE | 192 MSE | 192 MAE | 336 MSE | 336 MAE | 720 MSE | 720 MAE | **Avg MSE** | **Avg MAE** |
> |-------|--------|--------|---------|---------|---------|---------|---------|---------|-------------|-------------|
> | 0.00  | 0.4590 | 0.4740 | 0.5170  | 0.5100  | 0.5620  | 0.5330  | 0.5660  | 0.5520  | 0.5260      | 0.5172      |
> | 0.25  | 0.4540 | 0.4690 | 0.5120  | 0.5050  | 0.5570  | 0.5280  | 0.5610  | 0.5470  | 0.5210      | 0.5122      |
> | 0.50  | 0.4190 | 0.4340 | 0.4770  | 0.4700  | 0.5220  | 0.4930  | 0.5260  | 0.5120  | 0.4860      | 0.4773      |
> | 0.75  | 0.3900 | 0.4050 | 0.4440  | 0.4380  | 0.4940  | 0.4650  | 0.4990  | 0.4840  | 0.4567      | 0.4480      |
> | 1.00  | 0.3780 | 0.3930 | 0.4460  | 0.4390  | 0.4810  | 0.4520  | 0.4850  | 0.4710  | **0.4475**  | **0.4388**  |
>
> We found that alpha equals 1.00 and 0.75 performed best; in some experiments, alpha = 0.75 and alpha = 1.00 had comparable accuracy.
> When alpha = 0.5, the prediction accuracy started to deteriorate; and when alpha was less than 0.25, the prediction accuracy was the worst.
> This indicates that slightly changing the TIFO weights (alpha = 0.75) has little impact on prediction accuracy, and in some experiments, it has almost no effect.
> However, significantly reducing the TIFO weights, or removing TIFO altogether, significantly reduces the accuracy and robustness of the model on unseen test data.

---

> ### Author Response · Authors · 2025-11-23
>
> > Regarding to Frequency weighting modifies amplitude and may implicitly influence phase after inverse transformation, particularly when using finite-length windows. The paper should examine possible phase distortions and boundary effects, and clarify whether overlap-add or tapering techniques are employed during reconstruction.
>
> Thank you for the insightful comment.
> As we showed in the method section, TIFO applies a global DFT on the full input sequence of length (L), multiplies each complex coefficient by a real-valued weight (\lambda_k), and directly applies the inverse DFT.
> TIFO does not segment the signal into overlapping frames, so no overlap-add (OLA) or window tapering is involved. We have implemented several STFT-style reconstruction variants with different windows and OLA strategies and compared them against original version. We use ETTh1 dataset and iTransformer backbone here. We compare four reconstruction settings:
>
> * Global DFT (Rect.): Rectangular window, no overlap (global FFT over the entire sequence; this is our baseline).
> * STFT (Hann, 50% ovlp)**: Hann window, 50% overlap, standard OLA.
> * STFT ($\sqrt{\text{Hann}}$, 50% ovlp): $\sqrt{\text{Hann}}$ window, 50% overlap, weighted OLA (WOLA), satisfying the constant-overlap-add (COLA) condition.
> * STFT (Hann, 75% ovlp): Hann window, 75% overlap, standard OLA.
>
> | Settings | Run 0 MSE | Run 0 MAE | Run 1 MSE | Run 1 MAE | Run 2 MSE | Run 2 MAE | **Avg MSE** | **Avg MAE** |
> | :--- | :--- | :--- | :--- | :--- | :--- | :--- | :--- | :--- |
> | **Global DFT (Rect.)** | 0.3902 | 0.4068 | 0.3939 | 0.4105 | 0.3974 | 0.4140 | **0.3938** | **0.4104** |
> | **STFT (Hann, 50% ovlp)** | 0.5887 | 0.5148 | 0.5623 | 0.5053 | 0.5951 | 0.5152 | 0.5820 | 0.5118 |
> | **STFT ($\sqrt{\text{Hann}}$, 50% ovlp)** | 0.3999 | 0.4135 | 0.3985 | 0.4133 | 0.4032 | 0.4175 | 0.4005 | 0.4148 |
> | **STFT (Hann, 75% ovlp)** | 0.5872 | 0.5130 | 0.5627 | 0.5050 | 0.5944 | 0.5149 | 0.5814 | 0.5110 |
>
> As shown in the table above, the default **Global DFT** setting gives the best results, while **STFT** variants (Hann-based) clearly degrade the performance.
> The COLA-compliant **STFT ($\sqrt{\text{Hann}}$)** variant is slightly worse than the original method.
> In our Global DFT setting, we only rescale the frequency components and keep the same finite-length FFT boundary assumption as the backbone model.
> This design keeps the temporal structure exactly aligned with the backbone input and avoids additional interpolation across frame boundaries.
> In contrast, windowing plus STFT (OLA/WOLA) reshapes the signal locally and mixes neighboring frames, which can ignore mid-to-high frequency features, leading to the observed accuracy drop.

---

> ### Author Response · Authors · 2025-11-23
>
> > Q1 Regarding the theoretical guarantee for its improvement in forecasting accuracy.
>
> Thank you for the discussion. We would like to clarify that we do not explicitly claim a theoretical guarantee for improved forecasting performance. Instead, our argument is that frequency-domain modeling provides a principled and moderate way to better measure stationarity, thereby helping address distribution shift, with following theoretical considerations:
> * though we do not claim that TIFO has the capability to improve directly on forecasting accuracy, its orthogonalization procedure can indeed improve on the representation capability via its data-specific eigen-decomposition (Mercer's theorem).
> * the eigen-decomposition allows us to harness the power of orthogonal representation in the frequency space. This leads to decorrelate the dimensions such that the frequency magnitudes (vectors in the frequency space) are mutually independent and enabling the measurement of intrinsic characteristics of time series that are typically intractable in the time domain.
>
> It is hard to directly translate improvement on representation to improved forecasting accuracy. However, with the Mercer's theorem that permits an eigen-decomposition (orthogonalization) in the frequency space, we are guaranteed that better representation can be found, and very likely improved accuracy in practice.
>
> > Q2 How does TIFO handle multivariate inputs? Is there a joint reweighting mechanism across channels to capture cross-spectral dependencies, or are the weights learned independently per channel?
>
> TIFO applies FFT and learns frequnecy weights independently for each channel, without an explicit joint reweighting mechanism across channels.
> This channel-independent design is similar in spirit to the channel-independent setting in PatchTST [1].
> We agree that explicit cross-channel spectral modeling is a promising direction, and we have added such discussion in Conclusion.
>
> [1] Nie et al., A Time Series is Worth 64 Words: Long-term Forecasting with Transformers, ICLR2023

---

> > ### Comment · Reviewer_NcYR · 2025-11-26
> >
> > Thanks for the authors' response; most of my concerns have been addressed.
> >
> > However, regarding W1 and Q1, I think that since one of the major contributions of this work is to address the distribution shift problem in TSF, the authors should provide a detailed background with either experimental results showing the existence of such a phenomenon, or a review of prior works that emphasize the importance of this problem. Furthermore, the current theoretical parts don't guarantee better performance of TIFO, which makes the theoretical contribution seem incomplete.
> >
> > I will discuss with the other reviewers to adjust my final rating.

---

> > > ### Author Response · Authors · 2025-11-27
> > >
> > > Dear Reviewer NcYR,
> > >
> > > We are very glad to hear that our responses have addressed your concerns.
> > > Regarding the remaining points about W1 and Q1, we improved as follows:
> > >
> > > * (detailed background): We have already provided additional a *Addtional Discussion about Background Section* in Appendix B.1, along with illustrative figures (Fig. 4). In the revised version, we have further improved both the Introduction (Line 35–40, and 47-48) and the Background section (Line 756–780) to more clearly describe the distribution shift issue in time series forecasting.
> > > * (experimental results of the phenomeno existence): We have provided quantitative evidence in Section 4.2, *Frequency Domain Shift Analysis*, we explicitly quantified the spectral distribution shift between training and testing data. As shown in Table 3 and Fig. 2 of the manuscript. The results prove that significant spectral shifts exist and that standard normalization (RevIN) fails to eliminate shifts.
> > >     * We utilized Jensen-Shannon Divergence ($JSD^2$) to measure the overall distributional gap between training and test datasets.
> > >     * We used the Kolmogorov-Smirnov ($KS$) statistic to measure the maximum deviation.
> > > * (prior work review): We have revised Section 5 (*Related Work*) to discuss prior works that emphasize this problem, including a series of works for tackling this research topic. We also provided the differentiation to better position our work.
> > > * (performance guarantee): Our goal is to provide a mathematical framework showing that TIFO is not merely an empirical design, but is grounded in a principled way of modeling time-series distributional structures through frequency-domain operations. To the best of our knowledge, this is the first theoretical analysis that serves both the shift-problem formulation and the effectiveness of frequency-domain modeling. We hope to expand this direction in future exploration, and we hope that our analysis framework can contribute to and motivate further research in this direction.
> > >
> > > We sincerely appreciate your time, consideration, and constructive discussion throughout the review process. We kindly hope the reviewer will reconsider the rating.
> > >
> > > Authors.

---

> > > > ### Comment · Reviewer_NcYR · 2025-11-27
> > > >
> > > > Thanks for the detailed response and revisions.
> > > >
> > > > I'm raising my score.

---

> > > > > ### Author Response · Authors · 2025-11-27
> > > > >
> > > > > Dear Reviewer NcYR,
> > > > >
> > > > > We appreciate your confirmation that our responses have addressed your comments. Thank you very much for your time and your valuable contribution to improving our manuscript.
> > > > >
> > > > > Authors.

---

### Official Review · Reviewer_vcHc · 2025-10-30

**Soundness:** 3
**Presentation:** 3
**Contribution:** 2
**Rating:** 4
**Confidence:** 3

**Summary:**

The paper proposes TIFO (Time-Invariant Frequency Operator) for stationarity-aware representation learning in time-series forecasting. The key idea is to estimate dataset-level frequency stationarity and use it to reweight Fourier coefficients before feeding sequences into standard backbones (DLinear, PatchTST, iTransformer). Conceptually, the method argues that the DFT induces a frequency-space eigen-decomposition; learning data-specific eigenvalues (via small MLPs that output real/imaginary weights) emphasizes stationary components and suppresses non-stationary ones, thereby reducing train–test spectral shift and improving generalization. Empirically, the paper reports 18× top-1 and 6× top-2 wins across 28 settings, large gains on ETTm2, and 60~70% runtime reductions relative to certain baselines.

**Strengths:**

- This work focuses on the practical and important issue of *non-stationarity* in time-series forecasting, providing a frequency-domain perspective that complements conventional normalization approaches focused on low-order statistics..

- This work proposes a tightly integrated, plug-and-play pipeline: Stage-I computes frequency- and channel-wise stationarity $S(k,c)$​; Stage-II maps SSS through two lightweight MLPs to produce real/imag frequency weights that reweight DFT coefficients before iDFT, ensuring real-valued reconstructions.

- The proposed TIFO module is lightweight, generalizable, and can be easily integrated into diverse backbones (DLinear, PatchTST, iTransformer) without architectural modification.

**Weaknesses:**

(i) The conceptual novelty, while interesting, mainly lies in *reinterpreting normalization through frequency reweighting* rather than establishing a fundamentally new principle. The relation to prior spectral or stationarity-aware frameworks (e.g., FAN, FedFormer, FILM, FredFormer) is not deeply analyzed, leaving unclear how TIFO’s weighting differs from existing frequency-domain normalization or filtering approaches.

(ii) The paper underexplores sensitivity of key design choices: the definition/estimation of $S$ (mean/std vs. robust variants, windowing) and frequency resolution $K$. Existing ablations mainly randomize the initialization vector $s$, leaving robustness/controllability less clear.

(iii) The paper does not explicitly analyze how TIFO addresses the concrete shortcomings of prior approaches and why its design succeeds where they fail.  For example, beyond stating that time-domain normalization focuses on low-order statistics and may leave spectral shifts unresolved, and that frequency methods like top-k masking risk discarding informative peaks, the paper does not map TIFO’s dataset-level stationarity weighting and per-frequency learned coefficients to specific failure modes (e.g., residual train–test spectral shift after RevIN/SAN; over-suppression or mis-selection in FAN;  differences from FILM/FedFormer/FredFormer’s spectral handling).  Targeted comparisons or case studies (error breakdowns, where TIFO helps vs. hurts) would clarify the mechanism and strengthen the conceptual contribution.

**Questions:**

Please see 'weakness', which simply can be summarised as:

(i) Please elaborate on the core conceptual novelty of TIFO beyond reinterpreting normalization through frequency reweighting, and clarify how it fundamentally differs from existing spectral or stationarity-aware frameworks (e.g., FAN, FedFormer, FILM, FredFormer).

(ii) Please provide a systematic sensitivity analysis of key design and implementation choices, including the definition and estimation of $S$ (e.g., mean/std vs. robust variants, windowing strategy) and frequency resolution $K$​.

(iii) Please expand the Related Work or an analysis section to explicitly map how each TIFO component addresses the concrete shortcomings of prior approaches and substantiate these explanations with targeted comparisons or failure-mode case studies (e.g., where TIFO helps or fails).

---

> ### Author Response · Authors · 2025-11-23
>
> We thank the reviewer for the careful reading and feedback. Below we address each weakness and question with additional analysis and clarifications that we will incorporate into the revised version.
>
> >Regarding the core conceptual novelty of TIFO beyond reinterpreting normalization through frequency reweighting.
>
> Our analysis of normalization behaviors leads to two fundamental innovations:
>
> 1) a new modeling strategy, cross-sample stationary learning from the dataset (line 143 and 207, Stage-I), which can capture how data evolves globally across training and test domains. This is different from all existing sample-wise normalization methods (FAN, FILM, ReVIN) and architecture-based frequency normalization methods (FedFormer, FredFormer, etc.). Therefore, TIFO is concerned with "which frequencies remain stable across different samples over a long period under the training distribution."
> 2) a novel and effective non-stationary/stationary measuring and learning method that
>   2-1) estimates the global distribution of different frequencies across all samples by proposing a new metric,
>   2-2) learns sample-wise stationary features adaptively, which can prevent instance information loss.
> 3) Beyond empirical method in existing works, we provide theoretical analysis to further provide explain our method using kernel methods and spectral decomposition. We confirm TIFO provides a stability-based prior using an empirical eigenvalue estimate of the data-induced kernel.
>
> In short, we propose a new method in both modeling and methodology, rather than just an incremental weighting modification.
>
> >Regarding how our method fundamentally differs from existing spectral or stationarity-aware frameworks (e.g., FAN, FedFormer, FILM, FredFormer).
>
> * FedFormer, FILM, FredFormer: These frequency-learning and weighting methods (1) are architecture-based methods that cannot be extended to different forecasting models, and (2) do not focus on the non-stationary issue in time series. In contrast, our method is a plug-and-play module specifically designed to handle non-stationary data.
> * FAN is plug-and-play normalization method, but it is sample-wise approaches and relies on a heuristic top-$k$ masking strategy to zero out a subset of components based on amplitude.
>
> We have explicitly stated all above discussion in the Related Work section of the revised manuscript. Please kindly see in *Section 5*.

---

> ### Author Response · Authors · 2025-11-23
>
> >Regarding systematic sensitivity analysis of key design and implementation choices, including the definition and estimation of (e.g., mean/std vs. robust variants, windowing strategy) and frequency resolution.
>
> We thank the valuable comments and have added the new table and discussion in *Table 11, Appendix*.
>
> - Window Functions: rect (rectangular window), hann (Hann window)
> - DFT Lengths: 48, 96
> - adding mean/std to all experiments
>
> | Setting | Run 0 MSE | Run 0 MAE | Run 1 MSE | Run 1 MAE | Run 2 MSE | Run 2 MAE | **Avg MSE** | **Avg MAE** |
> |---------------|-----------|-----------|-----------|-----------|-----------|-----------|-------------|-------------|
> | rect_K48      | 0.4250    | 0.4420    | 0.4020    | 0.4180    | 0.4050    | 0.4210    | 0.4107  | 0.4270  |
> | rect_K96      | 0.3902    | 0.4068    | 0.3939    | 0.4105    | 0.3974    | 0.4140    | **0.3938**  | **0.4104**  |
> | hann_K48      | 0.4130    | 0.4260    | 0.4100    | 0.4240    | 0.4120    | 0.4250    | 0.4117      | 0.4250      |
> | hann_K96      | 0.4110    | 0.4240    | 0.4090    | 0.4230    | 0.4130    | 0.4260    | 0.4110      | 0.4243      |
>
>
> The results in the table show that when using the original data, retaining the complete spectrum can improve the prediction accuracy. When using the Hamming window, the frequency resolution has less impact on the forecasting accuracy, which may be because the Hamming window can reduce the frequency domain resolution of the signal. At the same time, the prediction accuracy is also reduced when using the Hamming window. The setup of experiments are as follows
> * with an input length of 96, we used DFT to transfer the time-domain data to the frequency domain. K96 means setting the frequency domain resolution to 96 frequency components (using all frequencies). K48 means retaining only the first 48 frequency components of the low-to-mid frequency range.
> * "rect" means using the original data, and "hann" means using the Hamming window function to process the data.

---

> ### Author Response · Authors · 2025-11-23
>
> >Regarding the Related Work or an analysis section to explicitly map how each TIFO component addresses the concrete shortcomings of prior approaches and substantiate these explanations with targeted comparisons or failure-mode case studies (e.g., where TIFO helps or fails).
>
> Thank you for your insightful comments. We have added a new Related Work section to the revised version, linking each TIFO component to the specific shortcomings of existing methods. Our core concern is how to address the challenge of train-test distribution shift in the frequency domain for non-stationary time series. Then, we categorize related work and TIFO components into three types:
>
> 1. Time-domain normalization (e.g., RevIN/SAN [1, 2]): These methods mitigate distribution drift by normalizing the time-domain mean and variance of each sample, but they do not explicitly observe the overall energy distribution of the training set in the frequency domain. Therefore, a significant residual train-test shift can still be observed in our frequency domain analysis.
> TIFO's dataset-level stationarity weighting (S(k,c)=\mu_{k,c}/\sigma_{k,c}) aims to address this issue: it calculates the cross-sample stability of each frequency-channel on the training set and uses this to guide subsequent frequency-domain reweighting, thus explicitly targeting the failure mode where the spectrum remains misaligned after RevIN/SAN.
>
> 2. Frequency domain Attention/Top-k Selection(e.g., FFT-based frequency domain modules, FAN-like methods[3,4,5,6,7]): These works typically perform attention or top-k masking on the spectrum of a single sample or a local window. Decisions are based on locally instantaneous high-energy frequencies, which can lead to inadvertently deleting stable but low-energy frequencies or retaining accidentally high-energy frequencies that are highly unstable at the dataset level.
> TIFO, based on (S(k,c)), learns weights (\lambda(k,c)) for all frequencies. With dataset-level stability as a priori, it performs continuous shrinkage on unstable frequencies instead of hard pruning, and shares this across all samples, thus specifically mitigating this type of over-suppression/mis-selection failure mode.
>
> 3. Embedded frequency domain models (such as FedFormer/FiLM/FredFormer  [4,5,6,7]): These methods move the operators of Transformer/MLP models to the frequency domain, primarily aiming to improve long-period modeling capabilities and efficiency. They represent an architecture-level frequency domain design, but do not focus on learning frequency stationarity.
> In contrast, TIFO is designed as a preprocessing layer decoupled from the backbone architecture. This is complementary to, rather than repetitive to, the architecture-level frequency domain module mentioned above.
>
> [1] Kim et al., Reversible Instance Normalization for Accurate Time-Series Forecasting against Distribution Shift, ICLR2022.
> [2] Liu et al., Adaptive Normalization for Non-stationary Time Series Forecasting: A Temporal Slice Perspective, NeurIPS2023.
> [3] Ye et al., Frequency Adaptive Normalization for Non-stationary Time Series Forecasting, NeurIPS2024.
> [4] Wu et al., TimesNet: Temporal 2D-Variation Modeling for General Time Series Analysis, ICLR2023.
> [5] Zhou et al., FEDformer: Frequency Enhanced Decomposed Transformer for Long-term Series Forecasting, ICML2022.
> [6] Zhou et al., FiLM: Frequency improved Legendre Memory Model for Long-term Time Series Forecasting, NeurIPS2022.
> [7] Piao et al., Fredformer: Frequency Debiased Transformer for Time Series Forecasting, KDD2024.

---

> > ### Comment · Reviewer_vcHc · 2025-11-24
> > **Response to the authors**
> >
> > I appreciate the authors’ detailed rebuttal, which has largely addressed my earlier concerns about the novelty and experimental validation of the work. In view of this, I have adjusted my score upward.

---

> ### Author Response · Authors · 2025-11-27
>
> Dear Reviewer vcHc,
>
> We are very glad to hear that our responses have addressed your concerns. We sincerely appreciate your time, consideration, and constructive discussion.
>
> Authors.

---

### Author Response · Authors · 2025-11-23
**Initial global response to all reviewers**

We sincerely thank all reviewers for their careful reading, constructive feedback, and thoughtful suggestions. We have revised the manuscript extensively in response. Below, we summarize the major updates:

(Reviewer NcYR) Revised Lines 189–194 to clarify the theoretical intuition.

(Reviewers vcHc, fb1q) Expanded the related works section with targeted comparisons against recent frameworks (e.g., RevIN, FAN, FITS) (see Section 5).

(Reviewer NcYR) Added ablation on robustness–expressiveness trade-off by scaling TIFO weights (Table 12, Appendix C.4).

(Reviewers NcYR, fb1q) Added ablation on reconstruction strategies (DFT vs. STFT/Overlap windowing) to address phase distortion and boundary effects (Table 13, Appendix C.4).

(Reviewers vcHc, Xjwr) Added ablation on window functions and FFT resolution to verify sensitivity (Table 11, 13, Appendix C.4).

(Reviewers Xjwr, fb1q) Added ablation on online EMA refresh to test robustness under distribution drift (Table 14, Appendix C.4).

(Reviewer Xjwr) Added ablation on STFT hyperparameters (Table 15, Appendix C.4).

(Reviewer Xjwr, fb1q) Added an ablation study and discussion in Appendix B.3.

---

### Author Response · Authors · 2025-11-28
**Thanks to All Reviewers**

Dear Reviewers,

We are very glad to hear that our responses have addressed your concerns.
Your discussion and valuable comments have greatly improved our manuscript. We sincerely appreciate your time, consideration, and constructive feedback.

Authors

---

### Author Response · Authors · 2025-11-29
**Paper and Rebuttal Summary**

Dear ACs,

We have addressed all reviewer's concerns during the rebuttal period, and **all reviewers have either raised or maintained a positive score**.
For confirmation, please kindly refer to **the end of each reviewer’s response section**.

We sincerely appreciate your time, consideration, and efforts.

Authors

---

### Meta-Review · Area_Chair_6k9H · 2026-01-03

**Summary:**

The submission introduces TIFO (Time-Invariant Frequency Operator), a plug-and-play preprocessing module designed to mitigate distribution shift in time-series forecasting by reweighting frequency components based on dataset-level stationarity.

While the reviewers found advantages in the empirical results, the integration into existing models, and the computational efficiency, there are significant concerns raised regarding the conceptual novelty of the work.

1. reviewers questioned whether re-interpreting normalization through frequency re-weighting is a fundamental novel contribution or an incremental contribution of existing frequency methods (e.g., FAN, FedFormer, FITS)

2. initial assessments highlighted a gap between the provided theoretical motivation and the actual model design.

Despite the authors' efforts during the rebuttal,  this paper lacks the depth of innovation required for ICLR. Therefore, I recommend a Reject.

**Reviewer Concerns:**

The authors successfully provided a systematic sensitivity analysis regarding FFT hyper-parameters. Other experience are also added. The authors clarified that TIFO addresses spectral shifts that persist even after time-domain normalization (like RevIN). The authors provided quantitative evidence through JSD and KS statistics.

While the authors distinguished TIFO from sample-wise methods, the core mechanism of it is applying a learnable weight to frequency components, which remains very similar to existing frequency methods. Reviewer fb1q noted that this diminishes the paper's novelty.

As noted by Reviewer NcYR, while the theoretical analysis using Mercer's Theorem provides a mathematical formulation for the decomposition, it does not provide a guarantee of performance improvement.

My additional comments are that this paper ignored Koopa (NeurIPS), which also discussed dataset-level frequency stationarity as I recalled. It makes the paper less novel.

**Reviewer Scores:**

All the reviewers would keep their original scores.

---

### Decision · Program_Chairs · 2026-01-26

Reject